# Tacrolimus dosing in liver transplant recipients using phenotypic personalized medicine: A phase 2 randomized clinical trial

Jeffrey Khong [1,9], Megan Lee[2,9], Curtis Warren[3,9], Un Bi Kim[3,9], Sergio Duarte[3], Kenneth A. Andreoni[4], Sunaina Shrestha[3], Mark W. Johnson[3], Narendra R. Battula[5], Danielle M. McKimmy[3], Thiago Beduschi[3], Ji-Hyun Lee [6,7], Derek M. Li[7], Chih-Ming Ho [8] & Ali Zarrinpar [3] ✉

Tacrolimus is the most commonly used immunosuppression drug after solid organ transplantation; however, its dosing is challenging due to substantial inter-individual variability, often resulting in blood levels that deviate from the target therapeutic range. We explored whether a dynamically customized, phenotypic-outcome-guided drug dosing method could improve maintenance of drug trough levels within pre-determined target ranges, focusing on tacrolimus immediately after liver transplantation. This single-center, partially blinded, completed clinical trial involved 62 adults undergoing liver transplantation, block randomized into parallel groups: standard-of-care (SOC) clinician-determined or Phenotypic Personalized Medicine (PPM)-guided tacrolimus dosing. The primary outcome was percentage of post-transplant days with large (>2 ng/mL) deviations from the target range. At trial completion, analysis found statistically significant improvement in the PPM group ($n = 27$): 24.2% of days showing large deviations compared to 38.4% in the SOC group ($n = 29$) (difference −14.2%, 95% CI: −26.7 to −1.5 %, $P = 0.029$) with no increase in adverse events. These results demonstrate that PPM-guided tacrolimus dosing more effectively maintains drug levels within the target range compared to SOC, suggesting a promising approach to improving drug dosing. The trial was registered at ClinicalTrials.gov with the identifier NCT03527238.

Solid organ transplantation is a life-saving procedure for patients with end-stage disease, but it introduces the persistent challenge of preventing the recipient's immune system from rejecting the transplanted organ. Post-transplant immunosuppressive drug therapy is essential to maintaining the delicate balance between suppressing the immune response to avoid graft rejection and preserving enough immune function to protect against infections and cancer. Tacrolimus, the most frequently prescribed immunosuppressant in this context[1-3], inhibits calcineurin, a crucial enzyme for T cell activation, thereby selectively preventing the immune-mediated rejection of the

[1]Department of Psychology, University of California, Los Angeles, CA, USA. [2]Department of Biochemistry, University of California, Los Angeles, CA, USA. [3]Department of Surgery, College of Medicine, University of Florida, Gainesville, FL, USA. [4]Department of Surgery, Thomas Jefferson University, Philadelphia, PA, USA. [5]Department of Surgery, College of Medicine, University of Oklahoma, Oklahoma City, OK, USA. [6]Department of Biostatistics, College of Medicine, University of Florida, Gainesville, FL, USA. [7]Division of Quantitative Sciences, University of Florida Health Cancer Center, University of Florida, Gainesville, FL, USA. [8]Department of Mechanical and Aerospace Engineering, Henry Samueli School of Engineering, University of California, Los Angeles, CA, USA. [9]These authors contributed equally: Jeffrey Khong, Megan Lee, Curtis Warren, Un Bi Kim. ✉e-mail: Ali.Zarrinpar@surgery.ufl.edu

transplanted liver while allowing broader immune defenses to remain functional[4].

However, tacrolimus' narrow therapeutic window and high inter- and intra-individual variability in dosing requirements, particularly across diverse patient populations, necessitate clinician-titrated dosing that often results in deviation from target ranges, particularly during the critical post-operative phase[5–7]. High intra-patient variability in blood tacrolimus levels may be associated with poor outcomes, including rejection and graft loss[8–12]. Appropriate dosing of tacrolimus during the first 2 weeks after liver transplantation has further been associated with a lower risk of graft loss[13]. Therefore, there is a clear need for personalized medicine to address post-transplant immuno-suppression by improving therapeutic consistency and augmenting clinical decision making. A robust procedure to achieve personalized dosing of tacrolimus and other post-operative drugs has not been available[14].

Post-transplant immunosuppression provides a challenging model to test any personalized medicine platform. Previous attempts at tacrolimus dosing have used genetics, pharmacokinetics, and other predictive models[15–18]. However, it has proven difficult to simultaneously account for inter- and intra-individual variability in treatment regimens using such approaches, much less to use them to dose patients on a day-to-day basis. These differences also lead to health disparities not solely attributable to access, economics, or adherence[19–21]. Tacrolimus is a substrate of cytochrome P450 and P-glycoprotein, proteins with variable activity in intestine and liver[5,22,23]. Its clearance is dependent on liver and kidney function, which can vary tremendously in the post-transplant setting[24]. Furthermore, transplant patients take multiple interacting medications. Current methods based on population-averaged pharmacodynamics, pharmacogenetics, or pharmacokinetics cannot respond adequately to this variability.

Previously, there was no established transfer function that could quantitatively bridge the gap between molecular-level inputs and system-level outcomes. As a result, clinical dose adjustment relied on a trial-and-error dose titration approach. We recognized that the fundamental scientific basis of therapy lies in the application of pharmaceutical agents to address perturbed biological systems, functioning within the science realm of complex systems[25,26]. Consequently, we employed an unconventional inductive methodology, distinct from the reductionist approach and supported by experimental evidence[25]. This approach led to the discovery of a quantitative transfer function, correlating drug-dose inputs with the phenotypic responses of a complex biological system[27]. Central to this methodology is the Phenotypic Response Surface (PRS) function, the quantitative transfer function, which led to Phenotypic Personalized Medicine (PPM). This approach shows great potential for individualizing clinical treatments to achieve specific phenotypic objectives through the administration of personalized drug doses, for example, to maintain desired tacrolimus trough levels (TTL) as tested in this study. The mechanism-agnostic PRS function has been validated in various in vitro and in vivo models, encompassing over thirty distinct diseases like organ transplantation, cancer, and infectious diseases[14,28–33].

In this trial, we conducted a prospective pilot study with 4 PPM-dosed and 4 standard-of-care (SOC) dosed patients[14]. Comparisons between TTLs over the course of treatment showed that PPM dosing markedly improved the management of patient drug levels compared to SOC patients, including average daily deviation (ADD) analyses, the frequency and magnitude of deviation from the target range, and the post-transplant length of stay (LOS) in the hospital. Large deviations (> 2.0 ng/mL) from trough target ranges were deemed especially important as they were correlates of possible adverse events (immunologic reactions for levels far below range and neuro- or nephrotoxicity for levels far above range). Given these preliminary findings, we sought to conduct a larger randomized prospective clinical trial to compare PPM dosing to SOC dosing in maintaining post-liver transplant patients within their clinically indicated target range. The primary objective was to assess whether PPM could improve the maintenance of tacrolimus levels within the pre-determined target range, thereby reducing the frequency of large deviations. We found that PPM-guided dosing significantly reduced the percentage of post-transplant days with large deviations from the target range, resulting in better drug level management. Additionally, exploratory analyses indicated benefits such as shorter median hospital stays and faster normalization of liver enzyme levels in the PPM group. These results suggest that a dynamically customized dosing method can enhance therapeutic outcomes in post-liver transplant care.

## Results
### Participant flow and recruitment
Between September 1, 2018, and June 4, 2020, all adults scheduled to undergo primary, redo liver, or simultaneous liver/kidney transplantation at an academic transplant center were screened for eligibility (Fig. 1).

62 subjects met criteria and were randomized; 31 were assigned SOC-dosing and 31 were assigned PPM-dosing. Two patients enrolled in the PPM group did not proceed to the intervention phase of the study as their transplant operation was aborted after randomization. Tacrolimus was discontinued for one patient in each group due to neurotoxicity within the first few days after transplant, but as their tacrolimus levels were below the target range, neurotoxicity was deemed not dose-related. Furthermore, MRI imaging revealed no evidence of tacrolimus neurotoxicity. Both patients were switched to cyclosporine, and thus, there were no evaluable TTLs within the study period. One patient assigned to PPM dosing did not have tacrolimus level measured on post-operative day 5 due to a phlebotomy/laboratory error. As this would affect dosing accuracy and the study outcome measures, this patient was excluded from the study. One patient assigned to SOC refused multiple tacrolimus doses in the early post-operative period and was therefore excluded from the study. The remaining 56 (29 SOC and 27 PPM) patients underwent whole liver deceased unrelated-donor transplantation, completed the study, and were discharged from the hospital after transplantation. A total of 347 dosing decisions were made according to SOC in the control group; 223 dosing recommendations were made using PPM. No PPM recommendations were overridden by the clinical team. The study ended when the planned number of subjects were enrolled.

### Baseline data
In the study population of 56 patients, the median age was 58 years (48–62); 32 patients (57%) were male. The median body-mass index (BMI) was 28 kg/m² (25.1–32.1). The etiology of liver disease was alcohol-related in 21 patients (37.5%), non-alcoholic steatohepatitis in 13 patients (23%), and hepatitis C in 10 patients (18%). Ten patients (18%) had hepatocellular carcinoma, 2 (4%) underwent a redo liver transplantation, and 8 (14%) underwent simultaneous liver/kidney transplantation. No significant differences arose by chance in the baseline characteristics of the two groups. (Table 1 and Supplementary Table 1)

### Numbers analyzed and outcomes
**Tacrolimus trough level (TTL).** After liver transplantation, the tacrolimus trough level (TTL) is used as a measure of immunosuppression. If the TTL is too low, there is an increased risk of transplant organ rejection, and if too high, there is an increased risk of drug toxicity. Usually, the clinically pre-designated target range is 8–10 ng/mL. Per clinical standard, TTL represents the concentration of tacrolimus in the blood just before the next dose, i.e., 12 hours after the previous dose. Tacrolimus is given at 8 am and 8 pm; TTL is measured daily just before 8 am.

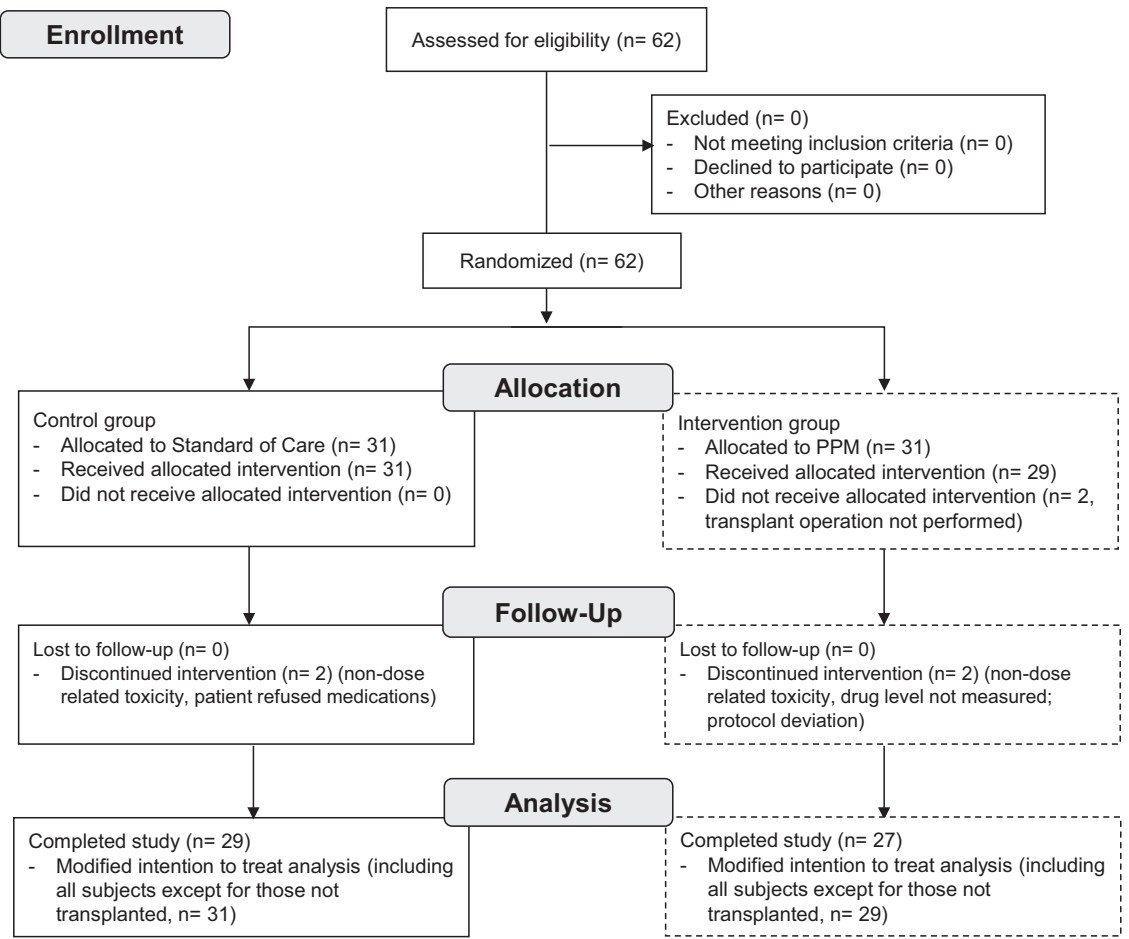

**Fig. 1 | CONSORT Flow Diagram Summarizing Participant Enrollment and Analysis.** The diagram shows the number of subjects assessed for eligibility, randomization, follow-up, and analysis stages, detailing exclusions, withdrawals, and final participant numbers analyzed. PPM: Phenotypic Personalized Medicine.

In our previous study, in the PRS function (Function 1), the phenotypic response is the desired tacrolimus trough level, TTL$(t+1)$ on day $(t+1)$, based on the previous doses, $c(t\text{-}3)$, $c(t\text{-}2)$, and $c(t\text{-}1)$, of a given patient to determine their personalized coefficients, $x_o(t)$, $x_1(t)$, and $y_{11}(t)$; these coefficients represent the phenotypic response to the drug dose. The tacrolimus dose on day $(t)$, $c(t)$, was administered to achieve the tacrolimus trough level for the patient on that day, i.e., the current day.

$$TTL(t+1) = x_0(t) + x_1(t)c(t) + y_{11}(t)c(t)^2 \qquad (1)$$

The PRS function dynamically predicts the personalized tacrolimus dosing regimen over a time course. It takes three days to determine the three coefficients to predict the tacrolimus dose of the fourth day. This is critical in the immediate post-transplant period when frequent changes to patient drug treatment regimens are common.

To increase the utility of PPM dosing, the trial design for the current study incorporated a much shorter equilibration period after liver transplantation compared to the pilot study (3 days versus 10 days, respectively). Due to this shorter equilibration period, there were larger fluctuations in patient physiology, including rapid changes in liver and kidney function, as well as a larger number of dose changes in immunosuppression and other drugs. The continuous three-day calibration procedure for Function 1 could therefore not be applied to this study. Inspired by Function 1 and having found that the next-day metabolism of tacrolimus closely reflects that of the current day—i.e., TTL$(t+1)$ / $c(t+1)$ = TTL$(t)$ / c$(t)$— the function for the tacrolimus dose

for next day, $c(t+1)$, was revised to Function 2:

$$c(t+1) = \left[ TTL(t+1)/TTL(t) \right] \times c(t) \qquad (2)$$

TTL$(t+1)$ is the desired next day TTL within the therapeutic window. This means that for dynamically optimized dosing of tacrolimus, only the individual's TTL and the tacrolimus dose from the previous day were needed to predict the tacrolimus dose for the next day. This approach does not incorporate any additional clinical variables, such as labs, donor age, or concomitant drug therapy, directly into the algorithm. The tacrolimus doses of PPM patients were therefore determined according to Function 2 to achieve the desired TTL within the clinically pre-designated target range[14].

**Primary outcome**

**Large deviations of TTL from the target range.** TTL higher than the target range may cause neuro- or nephrotoxicity, and lower than the target range may lead toward organ rejection. Hence, the primary outcome was the fraction of study dosed days (either SOC or PPM) during the initial post-transplant hospitalization with TTL deviating by more than 2 ng/ml from the target range, as this was deemed to be the closest correlation of dose-related adverse events. The first 3 days where tacrolimus in both groups was dosed according to SOC were not included in this analysis. Therefore, the initial post-transplant period was defined as from three days after the first tacrolimus dose post-liver transplantation until hospital discharge for both groups. In this study period, there was a significant difference in the mean percentage of post-transplant days with large deviations from the TTL between the

**Table 1 | Baseline characteristics of the analyzed study population**

|  | Standard of Care (n = 29) | PPM (n = 27) | P-value |
|---|---|---|---|
| Male | 16 (55%) | 16 (59%) | 0.75 |
| Female | 13 (45%) | 11 (41%) | – |
| HCC | 3 (10%) | 7 (26%) | 0.17 |
| SLKT | 5 (17%) | 3 (11%) | 0.71 |
| Redo OLT | 1 (3%) | 1 (4%) | 1.0 |
| Recipient Race/Ethnicity | – | – | 0.14 |
| Non-Hispanic White | 26 (90%) | 23 (85%) | – |
| Hispanic White | 1 (3%) | 4 (15%) | – |
| Non-Hispanic Black | 2 (6%) | 0 | – |
| Recipient Age (years) | 57 (41-61) | 58 (51–64) | 0.47 |
| BMI (kg/m$^2$) | 27.9 (25.4–37.5) | 28.6 (23.3–32.6) | 0.98 |
| NaMELD | 27 (18.5–30.5) | 27 (15–32) | 0.67 |
| Warm Ischemia Time (min) | 31 (26.5–38.5) | 33 (26–39) | 0.84 |
| Cold Ischemia Time (min) | 390 (310–440) | 330 (300–420) | 0.44 |
| DCD Donor | 0 (0%) | 1 (4%) | 0.48 |
| Donor Age (years) | 41 (25.5–58) | 38 (32–54) | 0.84 |
| Hepatitis C Positive Donor | 1 (3%) | 2 (7%) | 0.60 |
| Donor Risk Index | 1.38 (1.25–2.01) | 1.35 (1.07–1.66) | 0.18 |
| Dialysis After Transplant | 3 (10%) | 2 (7%) | 1.0 |
| Donor Cause of Death | – | – | 0.48 |
| Anoxia | 8 (28%) | 11 (41%) | – |
| Cerebrovascular Accident | 11 (38%) | 10 (37%) | – |
| Trauma | 10 (34%) | 6 (22.2%) | – |
| Fluconazole Use After Transplant | 17 (59%) | 16 (59%) | 1.0 |
| Mycophenolic Acid Use | 2 (7%) | 2 (7%) | 1.0 |
| Basiliximab Use | 10 (34%) | 7 (25.9%) | 0.57 |
| Tacrolimus Target Range Other Than 8–10 ng/mL | 4 (14%) | 5 (19%) | 0.72 |

HCC Hepatocellular Carcinoma; SLKT Simultaneous Liver-Kidney Transplant; OLT Orthotopic Liver Transplant; BMI Body Mass Index; NaMELD Sodium-Model for End-stage Liver Disease, DCD Donation after Circulatory Death. Categorical variables were compared between the two randomized groups using the chi-square test of independence, or Fisher's exact test when expected cell counts were less than five, with statistical significance assessed using a two-tailed test.

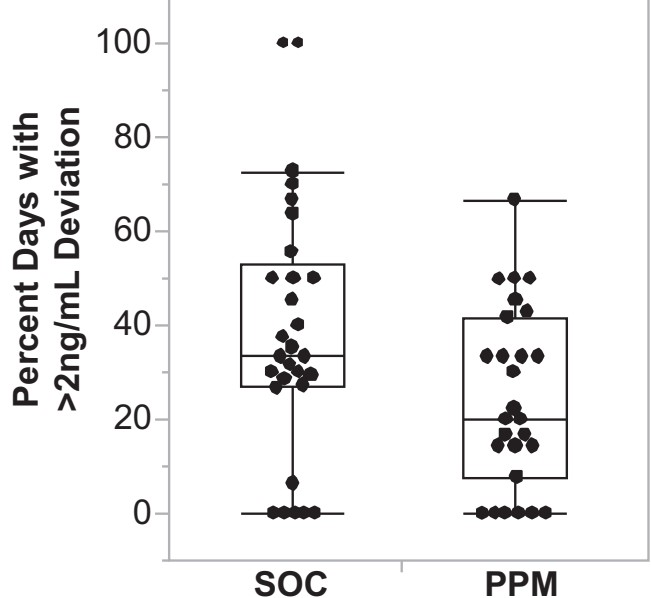

**Fig. 2 | Comparison of percent of dosed days with large deviations (> 2 ng/ml) from target range during the post-transplant hospitalization between standard-of-care (SOC, n = 29) and Phenotypic Personalized Medicine (PPM, n = 27) dosing groups.** PPM dosing resulted in a lower percent post-transplant days with large deviations from the target range compared to SOC dosing (P = 0.029). Data are presented as a box-and-whiskers plot, where the center line represents the median, the bounds of the box indicate the interquartile range (IQR, 25th to 75th percentile), and the whiskers extend to the minimum and maximum values within 1.5 times the IQR. A two-tailed Wilcoxon Rank Sum test was used to compare the groups.

two groups. The mean percentage (standard deviation) of post-transplant days with large deviations was 38.4 (27.4)% in the SOC group and 24.2 (19.1)% in the PPM group; (difference − 14.2%, 95% CI: − 26.7 to − 1.5%, P = 0.029) (Fig. 2). This difference remains statistically significant with a modified intention-to-treat analysis (including all subjects except for two randomized to the PPM group who did not undergo liver transplantation): 41.4 (29.2)% in the SOC group and 24.2 (18.8)% in the PPM group (difference − 17.2%, 95% CI: − 29.7 to − 4.6%, P = 0.0082). Furthermore, recognizing that the length of stay may differ between groups, we performed an additional analysis only up to 10 days post-transplantation. The results indicate a mean percentage (standard deviation) of post-transplant days with large deviations of 42.1 (30.9)% in the SOC group and 24.9 (19.5)% in the PPM group (difference − 17.2%, 95% CI: − 31.1 to − 3.5%, P = 0.015). Separately analyzing this outcome for subjects who did or did not receive a simultaneous kidney, or with or without HCC, does not change the overall findings; it only results in each of the groups not having enough subjects to reach statistical significance. (Supplementary Tables 2 and 3) A secondary outcome was percent of study dosed days with deviations

from the target range. Patients in the SOC group had a mean of 77.3 (14.2)% of post-transplant days with deviations from the target range; the PPM group had 71.7 (23.2)% of post-transplant days with deviations; (difference − 5.5%, 95% CI: 15% to 5.0%, P = 0.30).

## Harms
The incidence of biopsy-proven graft rejection episodes during the post-transplant hospitalization were similar (6 in each group; P = 0.89). There were no tacrolimus level-related episodes of nephrotoxicity or neurotoxicity, and there were no graft failures or patient deaths within the one-year follow-up of these patients. The incidence of neurotoxicity was 2 in the SOC group and 1 in the PPM group (P = 1.0), but none of these were found to be related to tacrolimus dosing, as the TTLs for all these cases were below the target range. As for nephrotoxicity, there were no biopsy-proven episodes of acute kidney injury or calcineurin toxicity or episodes of anuria or oliguria requiring dialysis. Overall, there were 3 subjects in the SOC group that required dialysis after transplantation and 2 in the PPM group (P = 1.0), but none of these were deemed to be related to too high TTLs; rather, these subjects had been on dialysis prior to transplantation and were clinically determined to need to continue on dialysis after transplantation.

## Exploratory outcomes
**Thirty-three percent earlier discharge of PPM patients.** Exploratory analysis revealed that patients in the PPM group had a 33.3% shorter median length-of-stay (LOS) compared to the SOC group. The PPM group had a median LOS of 10 (8–12) days, which is statistically significantly shorter than the SOC group median LOS of 15 (10.5–20.5) days (difference − 5, 95% CI: − 2 to − 8, P = 0.0026) (Fig. 3).

To quantify the degree to which the TTLs of each patient deviated from the target range, a secondary outcome is the average daily

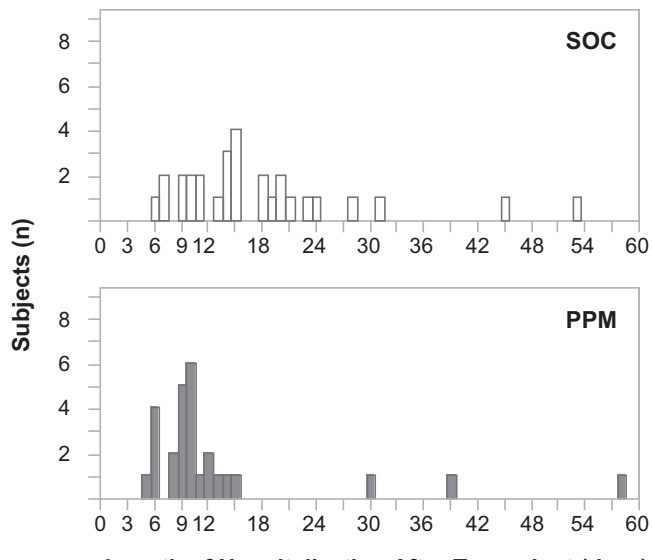

**Fig. 3 | Differences in Length of Hospitalization after Liver Transplantation.** Patients in the SOC group (*n* = 29) had a longer median length of stay (LOS) after transplantation of 15 (10.5–20.5) days; the PPM group (*n* = 27) had a median LOS of 10 (8–12) days (*P* = 0.0026). A two-tailed Wilcoxon Rank Sum test was used to compare the groups.

deviation outside-of-target-range (OOR) per day: ADD OOR = sum [TTL(OOR)] / study dosed days. SOC tacrolimus dosing led to an ADD of 1.81 (0.88) ng/mL/day; PPM dosing had a mean of 1.40 (0.79) ng/mL/day (difference − 0.41, 95% CI: − 0.85 to 0.03, *P* = 0.087). The difference in the ADD values between the SOC group and the PPM group was not statistically significant. However, the mean TTL of the PPM group averaged over the length of the study for each patient was higher than the SOC group (and within the commonly used target range of 8–10 ng/mL) [8.4 (1.4) ng/mL versus 7.6 (1.6), difference 0.8, 95% CI: 0.002 to 1.6, *P* = 0.034]. (Fig. 4) This difference in the mean TTLs prompted us to explore systematic differences between the SOC and PPM dosing methods.

We therefore tested whether either method was consistently under- or overdosing patients. As for overdosing, the ADD above the target range (i.e., TTL minus 10 ng/mL) is a measure of the chance of having drug toxicity. There was no significant difference in ADD above target range between the two groups [SOC: 0.42 (0.47) ng/mL/day versus PPM: 0.64 (0.58) ng/mL/day, (difference 0.22, 95% CI: − 0.07 to 0.50, *P* = 0.13)] (Fig. 5A).

With regard to underdosing (i.e., TTLs below target range), we found that the SOC group had a statistically significantly larger ADD below target range than the PPM group [1.4 (1.0) ng/mL/day versus 0.76 (0.51) ng/mL/day, respectively (difference − 0.64, 95% CI: − 1.05 to − 0.23, *P* = 0.0096)] (Fig. 5B). The ADD below target range (i.e., 8 ng/mL minus TTL) is a measure of the chance of suffering immunologic graft injury, consistent with previous findings correlating tacrolimus concentration and liver graft integrity markers[12].

**Early recovery of AST associated with shorter length of stay of PPM group.** To investigate the relationship between recovery from transplant-related liver injury and LOS, we analyzed the number of days it took for AST levels (a standard measure of liver injury) to reach a normal value, 48 U/L, referred to as Days to AST Normalization (DAST). The PPM group reached normal AST levels more quickly than the SOC group [PPM: 6 days (4-8); SOC: 8.5 days (6.25−11.75)]; *P* = 0.014 (Fig. 6). Six patients, three in each group, whose AST levels did not reach normal levels prior to discharge, were excluded from this

analysis. Including DAST values extending beyond discharge for these subjects did not alter the findings, with the PPM group normalizing faster than the SOC group [PPM: 6 days (5–9); SOC: 9 days (6.5–13)]; *P* = 0.013.

The SOC group exhibited a statistically significant increase in the ADD below the target range when compared to the PPM group. This metric serves as an indicator of the likelihood of experiencing more graft injury. Consequently, the PPM group demonstrated accelerated recovery in terms of liver function and AST levels, possibly resulting in a reduced LOS. The Days to ALT Normalization (DALT) between the two groups were also compared. DALT was also lower in the PPM group than in the SOC group, but, because many subjects did not reach normal ALT levels prior to hospital discharge, this difference did not reach statistical significance (12.7 days for SOC vs. 8.0 days for PPM, *P* = 0.15).

## Discussion

Tacrolimus dosing in the early period after liver transplantation is a challenging task that involves factors other than dose alone; it includes the types and doses of other medications, liver and kidney function, intrinsic metabolism, donor and recipient pharmacogenetics, and other factors. The important task of getting tacrolimus blood levels to the therapeutic range without under- or overdosing is managed daily by the clinical team. This team typically includes transplant pharmacists and physicians who together incorporate a combination of factors such as organ function, recovery of bowel function and absorption, and concomitant medications. This complex system is difficult to navigate and has not lent itself well to predictive protocols or input-output transfer functions.

Based on the science of complex-systems and an inductive approach analysis of the experimental evidence, we established the PRS function, which serves as the foundation of the PPM platform. The PPM platform entirely uses quantitative phenotypic data (TTL in this current example) to personalize treatment. As a result, it is applicable and valuable in systems of varying complexity, such as in this trial, where PPM navigated the complex biological, physiological, and clinical context of post-transplant tacrolimus dosing. The discovery of preservation of metabolism ratio from one day to the next, Function 2, represents a directly actionable approach towards improved immunosuppression therapy.

In this study, we found that PPM dosing outperformed SOC (i.e., clinician-determined) dosing by decreasing the percent of days with large deviations from the target trough level range in daily dosing of tacrolimus after liver and liver/kidney transplantation. While previous studies have explored predictive diagnostic and prognostic technologies, no prospective trials have assessed the ability to affect and measurably improve the routine clinical care of patients[34,35]. This study represents an in-human application in a randomized clinical trial. It confirmed the findings of the pilot study with a larger, independent, and more heterogeneous cohort, at a different institution, and with statistically significant differences[14].

The primary outcome of this study was the percentage of days with a large deviation from the target trough drug level range during the initial post-transplant hospitalization, which was decreased in the PPM group compared to the SOC group (Fig. 2). PPM also improved short-term clinical outcomes in that patients in the PPM dosing group were discharged 33.3% earlier than the SOC group (Fig. 3).

One potential explanation of how better dosing led to shorter LOS is hinted at by the additional finding that PPM patients had a faster time to normalization of AST (Fig. 6). Previous research has shown that tacrolimus trough concentrations in the short-term after liver transplantation are associated with graft integrity[12]. After transplantation, the immune system recognizes the graft as foreign, initiating an immune response, causing inflammation and damage. AST, an enzyme released from injured hepatocytes, becomes elevated during such a

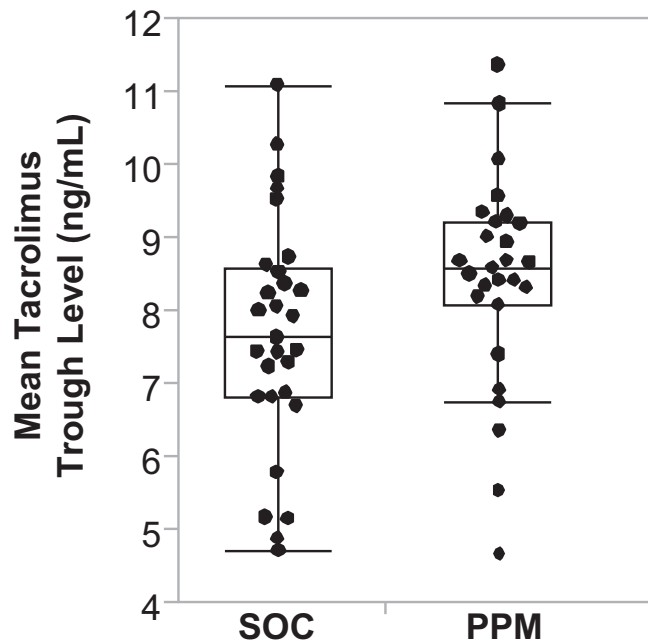

**Fig. 4 | Differences in TTL level after Liver Transplantation.** The mean tacrolimus trough level averaged over the length of the study for each patient in the SOC group ($n = 29$) was 7.6 (1.6) ng/mL; in the PPM group ($n = 27$) it was 8.4 (1.5) ng/mL ($P = 0.034$). Data are presented as a box-and-whiskers plot, where the center line represents the median, the bounds of the box indicate the interquartile range (IQR, 25th to 75th percentile), and the whiskers extend to the minimum and maximum values within 1.5 times the IQR. A two-tailed Wilcoxon Rank Sum test was used to compare the groups.

response. Tacrolimus and the other immunosuppressants inhibit the activation of T-lymphocytes and help prevent the immune-mediated damage. Appropriate immunosuppression levels, i.e., keeping patients within the higher in-range TTL using PPM (Fig. 4), help in the liver's recovery and function, as evidenced by a quicker return to normal AST levels. Not only is TTL higher in PPM-dosed patients overall, but SOC patients are underdosed more than PPM-dosed patients (Fig. 5B).

Preserving stable tacrolimus trough levels early after solid organ transplantation is a complex, yet crucial, task, as reducing the magnitude of trough level variability and the period of time in which large variations occur has a significant impact on post-transplant outcomes[12,36]. Multiple studies have found mean tacrolimus trough concentrations within the therapeutic range during the first 15 days after liver transplantation to be associated with a lower risk of graft loss when compared to under- and overdosing[12,13]. This was suggested to be linked with the significantly higher incidence of neurologic, cardiovascular, and acute renal failure complications in the high variability group. Furthermore, high intra-patient tacrolimus level variability in the first 30 post-transplant days has been correlated with significantly worse one-year and long-term graft survival[37]. Therefore, the decrease in the fraction of days with a large deviation from target levels is a clinically significant step towards improving graft and patient outcomes in liver transplantation. It is important to note that pharmacogenetic differences and socioeconomic factors can influence tacrolimus therapy outcomes in clinical practice. The PPM approach used in the trial is agnostic of genetics and socioeconomic background, relying on real-time therapeutic drug monitoring to optimize tacrolimus dosing, thus effectively managing variability in tacrolimus pharmacokinetics. Regarding cost, the PPM approach here uses data already collected as part of the standard of care. Furthermore, the processing and calculation required for PPM can be performed by a

trained transplant clinician in approximately the same amount of time as for standard dosing decisions. Therefore, the implementation of PPM does not result in significant additional costs or require extensive new resources.

This is a randomized prospective clinical trial and performed at an independent institution from the pilot trial[14], it is still limited by being a single-center, partially blinded study and its results will have to be replicated at multiple sites and in larger populations. In addition, since the randomization sequence was generated by the principal investigator using fixed block sizes to enable balanced group sizes, it may not have maintained complete concealment of allocation. Measures were taken to minimize bias, such as ensuring that subjects were blinded to group assignments and the clinical team was blinded to block size. How PPM may benefit the long-term patients' maintenance therapy, or its effectiveness in managing extended-release formulations of tacrolimus, will also need to be evaluated. This study was neither designed nor powered to detect differences in rejection or graft, or patient survival, as all of these outcomes are rare and depend on many variables other than dosing of tacrolimus. The potential limitations regarding the statistical analyses of secondary outcomes should be further acknowledged. For example, the pre-specified ADD analysis did not yield statistically significant results, whereas the exploratory analysis of mean TTL did show statistical significance. These findings may reflect type I errors due to multiple hypothesis testing. Readers should not over-interpret these exploratory findings, and future studies with larger sample sizes and appropriate adjustments for multiple testing will be needed to confirm these results.

In conclusion, the systematic application of drug dosing via PPM stands as a transformative approach in clinical care and offers dynamic, personalized recommendations for patients. As a result, the 33.3% early discharge rate among PPM patients signifies a clinically meaningful outcome, promising substantial benefits for patient well-being, fast recovery of liver function, and a notable reduction in healthcare costs.

## Methods

### Ethics/regulatory approval

The protocol was approved by the University of Florida Institutional Review Board (IRB201800053) to ensure the protecting the rights and welfare of participants in human subjects research, ensuring compliance with federal and state laws, local policies, and ethical principles, including those outlined in the Declaration of Helsinki. (Supplementary Note 1) All subjects provided written informed consent. A Data and Safety Monitoring Board reviewed the study six months after first patient enrollment and every six months thereafter. The study was to be stopped in the event of a significant unacceptable difference between the two groups, as determined by the Board.

### Trial registration

Optimizing Immunosuppression Drug Dosing Via Phenotypic Precision ClinicalTrials.gov (NCT03527238).

### Trial design and patient population

In this single-center, pragmatic, randomized, partially blinded trial, adult participants who underwent primary or redo liver or simultaneous liver/kidney transplantation at the University of Florida Health between September 1, 2018, and June 4, 2020, were assigned immediately prior to the transplant procedure 1:1 to daily standard-of-care physician-guided dosing or PPM-guided dosing of tacrolimus. The study period covered from the day of liver transplantation to the day of discharge. For the primary outcome, the first 3 days where tacrolimus in both groups was dosed according to SOC were not included in the analysis. Therefore, the initial post-transplant period was defined as from three days after the first tacrolimus dose post-liver transplantation until hospital discharge for both groups.

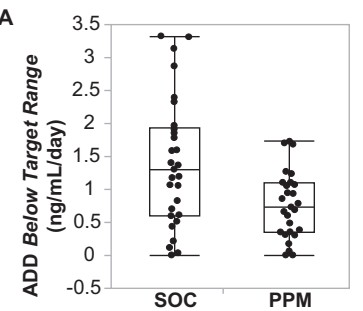
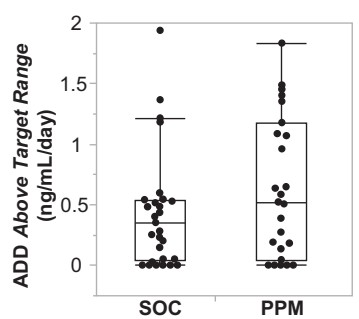
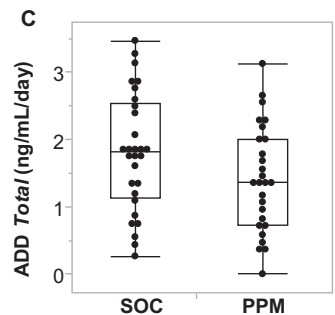

**Fig. 5 | Differences in Over- or Underdosing after Liver Transplantation.**
**A** Limiting the analysis only to *underdosing* after transplantation, the ADD below target range in the SOC group was 1.4 (1.0) ng/mL/day; in the PPM group, it was 0.76 (0.51) ng/mL/day ($P = 0.0096$). **B** Limiting the analysis only to *overdosing* after transplantation, the ADD above target range in the SOC group ($n = 29$) was 0.42 (0.47) ng/mL/day; in the PPM group ($n = 27$) it was 0.64 (0.58) ng/mL/day ($P = 0.13$). **C** Overall, the ADD in the SOC group was 1.81 (0.88) ng/mL/day; in the PPM group, it was 1.40 (0.79) ng/mL/day ($P = 0.087$). Data are presented as a box-and-whiskers plot, where the center line represents the median, the bounds of the box indicate the interquartile range (IQR, 25th to 75th percentile), and the whiskers extend to the minimum and maximum values within 1.5 times the IQR. A two-tailed Wilcoxon Rank Sum test was used to compare the groups.

No intervention was performed after discharge. Subjects were not compensated.

## Sample size and power calculations

The trial was planned based on previous results from the pilot study[14]. A sample size of 30 per group would achieve 92% power to reject the null hypothesis of equal means when the population mean difference of the primary outcome (percent days with large (> 2 ng/mL) deviation from target range) is 20% (31% vs. 11%) with standard deviations of 30% for SOC and 10% for PPM, and with a significance level of 0.05 using a two-tailed two-sample unequal-variance *t* test. The IRB approved the recruitment of an additional 15 subjects to account for study dropouts.

## Patient population

To ensure equal sample sizes, subjects were assigned into either the PPM group or the SOC group using a permuted block randomization with a block size of 4. Subjects were blinded to group assignments; the clinical team was not blinded. The clinical team was blinded to block size. The clinical team screened and attempted to enroll all eligible patients undergoing liver transplantation. The principal investigator generated the randomization sequence, and group assignment was assigned prior to the transplant operation to minimize bias from the operative course. To minimize dosing bias, the principal investigator was not involved in the dosing of SOC subjects.

Inclusion criteria: (1) Subject undergoing primary or redo liver or simultaneous liver/kidney transplantation; (2) Subject or surrogate able to provide informed consent; (3) Subject at least 18 years of age.

Exclusion criteria: (1) Enrollment in another investigational device or drug study; (2) Subjects compulsorily detained; (3) Psychiatric or medical illness that may put subject at significant risk, confound study results, or interfere significantly with subject participation in study; (4) Patients with contraindications to tacrolimus or pre-operatively anticipated to be switched off tacrolimus.

## Outcome measures

The primary outcome measure was percent days with large (> 2 ng/mL) deviation from the target range during the initial post-transplant hospital stay, as this was deemed to be the closest correlate of dose-related adverse events.

The secondary outcomes included percent days outside-of-target range, and the average daily deviation outside-of-target-range per day.

Safety outcomes included biopsy-proven graft rejection, graft failure, death, infections, nephrotoxicity (biopsy-proven acute kidney injury or calcineurin toxicity, anuria or oliguria requiring dialysis), or neurotoxicity (documented seizures, clinically significant tremors, or imaging-confirmed posterior reversible encephalopathy syndrome).

## Protocol and immunosuppression summary

Following liver transplantation at the University of Florida, all study patients were started on the SOC medication regimen per established center protocol:

- A three-drug combination of (i) oral tacrolimus (Prograf), (ii) corticosteroid (intravenous methylprednisolone 500 mg iv intra-operatively followed by intravenous methylprednisolone 250 mg iv on post-operative days 1 and 2, 125 mg on post-operative day 3 and oral prednisone taper starting on post-operative day 4 if able to tolerate oral medications), and (iii) mycophenolate [usually oral mycophenolate mofetil (CellCept) 500 mg q12h or oral mycophenolic acid (Myfortic) 360 mg q 12 h]. Induction was with intravenous methylprednisolone.
- Patients undergoing liver-kidney transplantation and those deemed at high risk for calcineurin-dependent renal injury also received basiliximab for induction and on post-operative day 4, starting tacrolimus on post-operative day 5-7 per clinical team discretion, and a lower tacrolimus target trough range.
- TTLs from a blood draw in the morning were measured using an automated chemiluminescent assay (CMIA; Abbott Laboratories) on the ARCHITECT i2000 platform. This was performed daily via until patient discharge.
- For the first 3 days that the patient was given tacrolimus, tacrolimus was dosed per SOC (usually capsule starting on day of transplantation at 0.05 mg/kg administered orally or sublingually every twelve hours) to allow enough data points to be gathered for PPM calculation.

  For SOC subjects:
- A clinical transplant pharmacist, together with the transplant-surgeon-on-call, continued to determine the subsequent doses.

  For PPM subjects:
- The daily treatment regimen details, including drugs already administered, drugs to be administered, and hemodialysis or any other procedures to be performed, were used for PPM analysis to provide the dosing recommendations.
- These data were sent to the PPM (prediction) team at UCLA. Following analysis, suggested tacrolimus doses were reviewed by a clinician and administered upon approval.
- The PPM process is not automated; PPM-suggested doses were based on safety limits set by the clinical team so that doses would always be within clinically-relevant levels.
- Patients remained on the trial until discharge from the hospital, at which point they reverted to SOC clinician-determined dosing. The standard target trough range was 8–10 ng/mL for the first month after transplant.

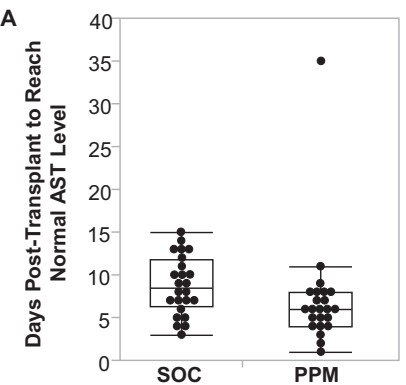
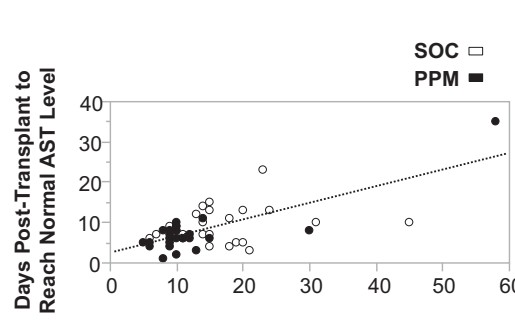

**Fig. 6 | Post-transplant Liver Enzyme Levels and Length of Stay. A** Patients in the SOC group (*n* = 24) took a median of 8.5 days (Q$_1$-Q$_3$ 6.25–11.75) to normalize aspartate aminotransferase (AST) levels. The PPM group (*n* = 23) reached normal AST levels at 6 days (Q$_1$-Q$_3$ 4–8) (*P* = 0.014). Data are presented as a box-and-whiskers plot, where the center line represents the median, the bounds of the box indicate the interquartile range (IQR, 25th to 75th percentile), and the whiskers extend to the minimum and maximum values within 1.5 times the IQR. A two-tailed Wilcoxon Rank Sum test was used to compare the groups. **B** Length of stay after transplantation was directly correlated with the number of days it took for AST levels to normalize (R$^2$ = 0.42, *P* = 10$^{-6}$), as assessed by a two-tailed Spearman rank correlation test.

- This 2 ng/mL range was adjusted based on the clinical condition of the patient per the clinical team and institutional SOC. Reasons to adjust the trough range included risk/concern/evidence for infection, rejection, malignancy, or neurological or renal dysfunction.
- The decision to discharge the subjects were made by the clinical team based on adequate pain control, mobility, and nutritional needs, therapeutic education, ongoing medical/surgical complications, and normal or normalizing liver function tests[38].

### Neuro- and nephrotoxicity assessment

- Nephrotoxicity was monitored through regular assessment of renal function, including serum creatinine levels, glomerular filtration rate (GFR), and urine output. Specifically, serum creatinine levels were measured daily during the initial post-transplant hospitalization. GFR was estimated using the Modified Diet in Renal Disease (MDRD) equation. Urine outputs were also recorded daily during the initial post-transplant hospital stay. These data were analyzed to identify any trends indicative of nephrotoxicity.
- Neurotoxicity was evaluated via clinical assessments for neurologic symptoms, including changes in mental status, seizures, or other neurological deficits. In addition, MRI imaging was utilized as appropriate for cases where clinical symptoms suggested possible neurotoxicity.

### Data processing

In this study, the PPM dosing was initiated for the fourth day of tacrolimus administration, which is earlier than in the previous pilot study[14]. Due to the shorter timeframe, there were larger fluctuations in patient physiology, including rapid changes in liver and kidney function. Additionally, the number of dose changes in immunosuppression and other drugs prevents a continuous three-day calibration procedure needed to apply Function 1. We had found that the next-day metabolism of tacrolimus is very close to that of the current day (Function 2). This means that for dynamically optimized dosing of tacrolimus, only the individual's TTL and the tacrolimus dose from the previous day were needed to predict the tacrolimus dose for the next day. The tacrolimus doses of PPM patients were therefore modulated according to Function 2 to achieve the desired TTL within the clinically pre-designated target range[14].

### Statistical analyses

Demographic and clinical characteristics were summarized by treatment group using descriptive statistics, including mean (standard deviation) for normally distributed data and median (first – third quartile, i.e., Q$_1$-Q$_3$) for non-normally distributed data. The analysis population included all subjects who received the allocated intervention (tacrolimus dosing using PPM versus SOC starting from day 4 of tacrolimus administration). The primary outcome measure was percent days with large (> 2 ng/mL) deviation from the target range during the initial post-transplant hospital stay. The secondary outcomes included percent days outside-of-target range and the average daily deviation outside-of-target-range per day.

Safety outcomes included biopsy-proven graft rejection, graft failure, death, infections, nephrotoxicity (biopsy-proven acute kidney injury or calcineurin toxicity, anuria or oliguria requiring dialysis), or neurotoxicity (documented seizures, clinically significant tremors, or imaging-confirmed posterior reversible encephalopathy syndrome). LOS, DAST, and ADD above and below the target range were assessed as exploratory outcomes after study completion. The primary analysis was performed on participants who completed the study as planned. In addition, a modified intention-to-treat (mITT) analysis was conducted as a secondary analysis to assess the robustness of the findings. The mITT population included all randomized participants based on their initial group assignments. However, subjects who did not undergo liver transplantation after randomization due to the cases being aborted could not be included in the ITT analysis as they did not have any measured TTLs.

Comparisons between the two groups were performed using either two-tailed Welch's *t* test or Student's *t* test for normally distributed data. For non-normally distributed data, a two-tailed Wilcoxon Rank Sum test was used to compare medians. Spearman's correlation was used to investigate the association between LOS and DAST. A significance level of 0.05 was used for all hypothesis tests. Statistical analyses were performed on JMP Pro 16.1.0 and confirmed with R 3.5.1.

### Reporting summary

Further information on research design is available in the Nature Portfolio Reporting Summary linked to this article.

## Data availability

Deidentified individual participant data underlying the results reported in this article, the complete study protocol, statistical analysis plan, and source data files for all graphs presented in the figures are included as supplementary files with the publication. Source data are provided in this paper.

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

## Acknowledgements

We thank the University of Florida Health Abdominal Transplant Program for their assistance in the conduct of this trial and for the care of the patients. This study was funded by National Institutes of Health/National Institute of Diabetes and Digestive and Kidney Diseases grant R21 DK116140 (A.Z. and C.M.H.); Robert Duggan-UF Foundation (A.Z.); UCLA-Robert Benson Funds R47441 (C.M.H.). The sponsors/funders had no role in study design, data collection, and analysis, or manuscript writing.

## Author contributions

Conceptualization: A.Z. and C.M.H. Methodology: A.Z. and C.M.H. Investigation, data acquisition, analysis, and interpretation: A.Z., U.B.K., C.W., S.S., K.A.A., M.W.J., N.R.B., S.D., T.B., and D.M.M. Development and performance of the PPM analysis, dose recommendations: C.M.H., J.K., M.L., and A.Z. Access and verification of the data, statistical analysis: J.K., M.L., J.H.L., D.M.L., C.M.H., and A.Z. Funding acquisition: A.Z. and C.M.H. Project administration: A.Z. and C.M.H. Supervision: A.Z. and C.M.H. Writing – original draft: A.Z., S.D., and C.M.H. Writing – review & editing:

J.K., M.L., C.W., U.B.K., S.D., K.A.A., S.S., M.W.J., N.R.B., D.M.M., T.B., J.H.L., D.M.L., C.M.H., and A.Z.

## Competing interests

C.M.H. is an inventor on pending and issued patents (International Patent Application Serial No. PCT/US2014/012111 and PCT/US2015/058892). CMH is a co-inventor of the pending patent WO2015017449. CMH and AZ are co-inventors of the issued patent US2019/0121935A1. C.M.H., A.Z., J.K., and M.L. are co-inventors of the pending patent (63/234,124). The remaining authors declare no competing interests.
