## [Peer Review File · Nature Communications]

Tacrolimus Dosing in Liver Transplant Recipients Using Phenotypic Personalized Medicine: A Phase 2 Randomized Clinical Trial

Corresponding Author: Dr Ali Zarrinpar

Version 0:

Reviewer comments:

Reviewer #1

(Remarks to the Author)

The manuscript titled "[Enhancing Tacrolimus Dosing Precision in Liver Transplantation A Phenotypic Personalized Medicine Approach]" is an important contribution to the field of tacrolimus therapy for transplant patients, as it explores the use of phenotypic personalized medicine to optimize dosing and reduce the risk of toxicity or lower efficacy. The authors propose an easy-to-apply approach that has the potential to significantly improve outcomes for transplant recipients requiring tacrolimus immunosuppression. However, the manuscript would benefit from clarification of various statistical methods, more precise terminology, and correction of some inappropriate or unclear statements as follows (ordered based on the first appearance in the text not by significance):

1- Line 111: Was the number of patients with simultaneous liver and kidney transplantation higher in one arm than the other?

2- Line 161: Authors mentioned "TTL higher than the target range may cause neural toxicity and lower than the target range may lead toward organ rejection" This is not accurate as Tacrolimus toxicity is not only neurological but can be also in the form of acute nephrotoxicity.

3- Line 166: Authors mentioned "There was a significant difference between the two groups" A significant difference in what?

4- Line 167: As the authors are providing numbers between brackets after the mean values, the statement "The mean percentage of post-transplant days with large deviations was 38.4 (27.4)%" needs to be changed to The mean percentage (standard deviation).

5- Line 168: Difference between 38.4-24.2 is 14.2% not 14.1%

6- Lines 173-175: Authors mentioned "There were no tacrolimus level related episodes of nephrotoxicity or neurotoxicity and there were no graft failures or patient deaths within the one-year follow up of these patients." How these toxicities were assessed. Have you followed up the cases with serum creatinine levels, GFR, and urine outputs. Any statistical data for this? & For neurotoxicity have you followed this up with MRI?

7- Line 178: authors mentioned post-hoc test. What was the posthoc test used herein. Have you performed a normality test? If yes, results for the normality test are not provided.

8-Line 184-185: Authors wrote this equation " $\text{Mean AUC OOR} = \frac{\text{sum [TTL(OOR)]}}{\text{study dosed days}}$ " and the units of this AUC was ng/mL/day.

The equation you provided for calculating the AUC (area under the curve) outside of the therapeutic range needs to be rechecked as it does not accurately capture the concept of AUC.

AUC typically represents the total drug exposure over time, calculated as the integral of the concentration-time curve. When considering AUC outside of a specified therapeutic range, the focus is on the drug concentrations that fall outside this range during the study period.

Sum of Trough Levels:

If you are summing trough levels (the lowest concentration of the drug in the bloodstream before the next dose), this may not accurately represent the AUC, which typically involves integrating all concentration values over time, not just trough levels.

Study Dosed Days:

Dividing by the number of study dosed days may provide an average concentration, but this does not equate to AUC. AUC is generally calculated as the area under the entire concentration-time curve, which requires more than just the trough levels. To accurately calculate the AUC outside of the therapeutic range, consider the following steps:

Identify Concentration Levels: Collect all concentration data points (not just troughs) that fall outside the therapeutic range during the study period.

Calculate AUC: Use the trapezoidal rule or another integration method to calculate the AUC based on these concentration values over time. This will provide a more accurate representation of total exposure.

Revised Equation

A more accurate representation could be:

AUC outside of the range

$= \int_{t_0}^{t_n} C(t) \cdot dt$

Where

C(t) represents the concentration values outside the therapeutic range over the study period.

Conclusion

For a precise calculation, consider integrating all relevant concentration data over time, rather than relying solely on trough levels.

9- Line 185: ng/mL/day unit is not a unit of AUC and it is not similar to ng·day/mL. They are different units with different meanings:

- ng/mL/day: This represents a rate of change in concentration per unit of time

It would be interpreted as nanograms per milliliter per day. For example, a drug concentration increasing by 10 ng/mL each day. This unit does not represent an area under the curve (AUC)

- ng·day/mL:

This represents an area under the concentration-time curve (AUC). It would be interpreted as nanogram-days per milliliter. It is a valid unit for expressing AUC when concentration is in ng/mL

The time component (days) is multiplied by the concentration. Integrating this over the dosing interval gives the total exposure as explained above.

10- Figure 3 needs clarification. At what timepoint of the follow-up were these TTL data collected?

11- Line 196: The title "Early Recovery of AST Associated with Shorter Length of Stay of PPM Group" is not relevant to the text below it

12- Line 210: The abbreviation (DAST) does not make sense with the description before it. Please check

13- Line 210: Authors mentioned that AST is used as it is the standard measure for liver injury. However, in clinical practice, alanine aminotransferase (ALT) is often prioritized when evaluating liver function and injury and is generally considered a better standard than aspartate aminotransferase (AST). Here are the key points regarding their roles and significance:

ALT vs. AST in Liver Injury

ALT Specificity: ALT is primarily found in the liver and is more specific to hepatocellular injury. Elevated levels of ALT are typically associated with different liver conditions such as viral, alcoholic, and non-alcoholic liver disease. In most liver injuries, ALT levels rise more significantly than AST levels, making it a more reliable marker for liver damage.

AST Sources: While AST is also present in the liver, it is found in other tissues, including the heart and muscles. Therefore, elevated AST levels can arise from non-liver sources, such as muscle injury or cardiac events, which can complicate the interpretation of liver function tests.

14- Have you considered other biochemical tests such as: Serum Creatinine, or better Cystatin C (for kidney function), Proteinuria, Electrolytes, Donor-Specific Antibodies (DSA; immunological marker for rejection), Neutrophil Gelatinase-Associated Lipocalin (NGAL), Cell-Free DNA, or Urinary Chemokines such as CXCL9 and CXCL10?

15- Lines 247-248: what was the duration of follow-up for the primary outcome?

16- Tables S1 & S2: Black subjects are more in the SOC than PPM that can contribute to the differences between the two groups apart from the personalized medicine initiative. Metabolism by CYP3A5 is known to be higher in black race compared to whites.

Additionally, the prognosis of tacrolimus therapy can be worse in Black patients compared to White patients, primarily due to pharmacokinetic differences, dosing requirements, and potential socioeconomic factors. Here are the key points regarding this disparity:

- Pharmacokinetics: Studies indicate that Black kidney transplant recipients often have a faster clearance rate of tacrolimus compared to White recipients. This means they may require higher doses to achieve therapeutic levels, which can complicate management and increase the risk of both toxicity and rejection if not properly monitored.

- Dosing Differences: Black patients typically receive nearly 55% higher doses of tacrolimus than their White counterparts to achieve similar drug exposure levels. Despite higher dosing, achieving target concentrations can still be challenging, leading to a higher incidence of underexposure (i.e., lower than target trough levels) in Black patients which we see in SOC group.

- Adverse Effects: There is evidence that Black patients may experience more adverse effects from tacrolimus therapy, which can further complicate treatment outcomes. The cumulative effects, particularly in Black women, have been noted to be more pronounced.

- Chronic Allograft Survival: Research has shown that chronic renal allograft survival tends to be shorter in Black recipients compared to other racial groups receiving similar immunosuppressive regimens. Factors contributing to this disparity include medication adherence, genetic differences in drug metabolism (particularly related to the CYP3A5 enzyme), and socioeconomic factors that can affect access to healthcare and medication management.

- Socioeconomic Factors: Disparities in healthcare access, socioeconomic status, and potential differences in health literacy may also contribute to the poorer prognosis observed in Black patients. These factors can influence medication adherence and overall health management, impacting transplant outcomes.

Reviewer #2

(Remarks to the Author)

In this single-center, pragmatic, randomized, partially blinded trial, liver transplant patients were randomized to undergo daily

standard-of-care (SOC) physician-guided dosing or Phenotypic Personalized Medicine-PPM guided dosing of tacrolimus (PPM platform uses quantitative phenotypic data -Tacrolimus trough levels-TTL in the prior 3 days) to personalize treatment. The main end point was the percent days with large (>2 ng/mL) deviation from target range during the initial post-transplant hospital stay. The main finding was the statistically significant improvement in the PPM group: 24.3% of days showing large deviations compared to 38.4% in the SOC group (difference -14.1%, 95% CI: -26.7 to -1.5 %, P=0.029).

Ensuring therapeutic target levels is very important for transplant recipients yet a simple automated way has been developed yet due the enormous complexity of the LT system and tac pharmacokinetics. Any improvement in that sense is welcomed.

There are some aspects that need to be addressed:

- The study was randomized yet the minimum number of patients that needed to be reached to achieve conclusive evidence (30 per arm) was not reached once some patients were discarded from the analysis. 27 and 29 patients in each arm. Were all consecutive patients included in the study in this "large transplant center" during a 1.9 years period? How many patients were not included during that period and what were the reasons?

- The study was not completely blinded such that subtle changes in the management of the SOC or PPM group can not be discarded. Were the physicians in charge of the patients the same during the study period? what was the overall LT experience of these physicians? was it similar for both groups?

- The authors state that the end-point was the % of large deviations during the initial post-transplant period? What do they mean by Initial? is initial the same by group? if LOS different in each group, the number of measurements are not the same by group and this may lead to bias. Were the number of measurements performed by group similar?

- The authors state: "Large deviations (> 2.0 ng/mL) from trough target ranges were deemed especially important as they were correlates of possible adverse events". Why was 2 and not 1 or 2 consider as relevant. Clearly this relates to the baseline /previous trough level. Changing from 1.5 ng/ml to 3.6 has clearly not the same meaning as changing from 10 to 12.1 or from 4 to 1.9. Time from LT is also important, situation of the patient, concomitant IS therapy. All are very relevant aspects that need to be considered instead of applying a "specific 8-10 range for all?". and this range may even change depending on the condition of the patient, particularly development of serious infections. How is this incorporated in the equation?

- The authors state that "No significant differences arose by chance in the baseline characteristics of the two groups. ". Yet some % are quite different. For instance, 26% of PPM group was HCC vs 10% in SOC group. Generally speaking HCC patients are easier to manage and target levels are more easily and rapidly achieved. Given the small number of patients included in this study, these differences between groups may have had an impact on the overall result.

- The explanation of the PPM system could be improved. Does that system only consider TTL during the previous 3 days or does it incorporate other variables such as level of ALT elevation, or donor age?

- While there was a statistical difference in the number of days showing large deviations, there were no differences in mean AUC above target range between the two groups. And while there were differences related to underdosing (SOC group had a statistically significantly larger mean AUC below target range than the PPM group), this did not impact rejection episodes or graft loss?

In fact, the differences using this PPM approach did not translate into clinical differences (same rate of rejection, graft loss, episodes of neuro or nephrotoxicity).

In essence, statistical differences but clinically relevant? In those with large deviations, it is important to understand the degree of deviation ? What was the mean deviation in one group over another? How many patients did it involve? given the small sample size a more detailed analysis would be welcomed.

Other comments

Table S2: the differences between the two randomized groups are shown for some but not all variables. For instance, HCC? MELD? Donor?

Please add some comments related to the lack of data when using prolonged release TAC, used in many LT centers in the world, which has shown better clinical results than standard twice daily tac.

The authors state that "Appropriate immunosuppression levels, i.e., keeping patients within the higher in-range TTL using PPM (Figure 3), help in the liver's recovery and function, as evidenced by a quicker return to normal AST levels". Yet it could be the opposite: a better and more rapid return to NO liver damage leads to lesser TAC variability.

Is there data from the kidney transplant setting? some patients were both Liver and Kidney transplant recipients? I understand the numbers are small, but what were the specific results in that subgroup?

Reviewer #3

(Remarks to the Author)

This article is to describe the conduct and resulting data and outcomes of a randomised trial to compare a novel

personalised method for deciding dosing of tacrolimus following liver and/or kidney transplantation compared to standard of care.

Overall it provides a good succinct description of the RCT performed. There are a number of recommendations which could improve the readers understanding of the research and evaluate it's likely value and impact.

Comment, line 128 states "study ended when the planned number of subjects were enrolled and completed the study" I suggest removing the latter part or rephrasing as the statement isn't quite right. The planned size was 30 per group, which although this number was enrolled, they didn't all complete the trial satisfactorily as stated in the paragraph above.

Table 1: It seems a little unusual to only have screened the exact number that were found eligible and consented to randomisation in the trial. In a consort flow diagram you would expect the study team to have screened and approached several more patients than those that ended up being included as you find ineligible patients or on explaining the study to the patient they fail to provide their consent or are reluctant to take part in research and therefore are never included. The number screened in this case is the same as those randomised. Furthermore in line 110, the authors state that "Between September 1, 2018, and June 4, 2020, all adults scheduled to undergo primary, or redo liver or simultaneous liver/kidney transplantation at a large academic transplant center were screened for eligibility." again suggesting that more patients might have been screened and not all would be expected to take part. It is important to describe screen failures to evaluate possible selection bias in the recruitment strategy as a single site study.

Randomisation: It should perhaps be acknowledged that the randomisation sequence was generated by the PI using fixed block sizes and therefore cannot accurately be considered independent or be seen to maintain concealment of allocation as is customary in RCTs.

Analysis plan: There is no mention of an analysis plan detailing how withdrawals or losses of data would be dealt with prior to data lock and unblinding of allocation in order to conduct analyses. Not including all randomised participants in a true intention to treat principled analysis plan may have led to the introduction of bias. An ITT analysis was reported as a secondary analysis, however there isn't a methods section describing how this analysis population was defined or derived making it hard to judge the importance/significance.

Line 37: refers to no significant differences between groups, these tables are presented as supplemental tables. Could the authors quantify how this assertion was made, did statistical tests confirm this, if so these results should appear in the S1 and S2 tables.

Line 166 simply states "There was a significant difference between the two groups." there is no indication what difference this is referring to (the previous or next sentence for example) and should be stated clearly and precisely within that sentence or removed completely.

The result of the secondary outcomes, fraction of days outside of target range doesn't appear to be reported at all and should included for transparency since it was an endpoint stated in the protocol.

Line 173 states no safety endpoints of nephrotoxicity and neurotoxicity related to tacrolimus were observed. Maybe it can be clarified if these events were observed but not considered related to tacrolimus, or that no events occurred at all in either arm? If the former the number and difference between groups should be reported in each group since relatedness could be subjective in an open label study.

The analysis methods of the exploratory DAST endpoint have not been stated, on line 212 it states "Six patients, three in each group, whose AST levels did not reach normal levels prior to discharge, were excluded from this analysis." suggests that the DAST variable was considered as a binary outcome rather than time to event data that could have included the 6 participants that were removed from the analysis and therefore be considered more appropriate for comparisons purposes.

The one sentence summary at the beginning of the article should perhaps reflect the primary endpoint results (as a minimum) in addition to or instead of the post hoc analysis of length-of-stay reported.

Overall the article would benefit from a methods section defining elements (as a minimum) such as how the different analysis populations were derived, the outcome measures analysed including how they were derived (especially those not stated up front in the protocol) and the analysis methods and test statistic used for each hypothesis test carried out and presented stating significance level and if 1 or 2-sided.

The discussion was sensible and outlined the main findings with the exception of the contradiction in results in terms of TTL outside of range. The stated secondary endpoint of fraction of days outside of target range was not presented. Furthermore, there were some inconsistencies in the results with the planned AUC OOR endpoint analysis providing a 'non-statistically-significant' result (p-value) whereas the apparent post-hoc analysis (i.e. not specified in the protocol a priori) of mean TTL did provide a 'statistically-significant' result (p-value). Perhaps the discussion might acknowledge this and discuss the potential for type I errors (false positive p-values) given the large number of hypothesis testing, post hoc analyses and apparent lack of adjustment for multiple testing.

Version 1:

Reviewer comments:

Reviewer #2

(Remarks to the Author)

The authors have adequately addressed all comments.

One additional question that was not addressed is the cost: is it different?

Reviewer #3

(Remarks to the Author)

I thank the authors for their thorough examination of the previous comments and for clear and concise responses and amendments to the manuscript.

There is one remaining point to be considered, in response to reviewer #3 point 4, the authors state:

“The primary analysis was performed on participants who completed the study as planned. TTLs that were not study dosed were excluded from analysis. Additionally, we performed, an ITT analysis as a secondary analysis to assess the robustness of the findings. The ITT population included all randomized participants based on their initial group assignments. However, there were subjects, such as those who did not undergo liver transplantation after randomization due to the cases being aborted (n=2), who could not be included in the ITT analysis as they did not have any measured TTLs. We have slightly adjusted the description of these 2 patients in the enrollment description to make it clearer (Page 7, Line 116-127). We have also revised the text in the Methods Section (Page 22, Line 382-388) to reflect this.”

The additions to the text are welcome and go some way to addressing the comment, thank you. Please note the typo in the additional text provided (page 22, line 387) the word ‘who’ is repeated and should be removed.

Use of the phrase ‘intention to treat’ is perhaps still misleading since it implies all participants are included in the analysis (true meaning of the phrase) whereas the authors clearly exclude some participants due to missing data. This approach might be more accurately described as a ‘modified intention to treat approach’. It is a limitation of the methods used that a true ITT population hasn’t been analysed whereby imputation for missing data could have enabled inclusion of all randomised participants.

I would suggest the authors ideally include the number of participants analysed in each of the analysis populations ‘completed study’ and ‘ITT’ in the consort diagram. Instead of the generic ‘analysed n=x’ it would be more transparent to list each analysis population and state the n that was analysed.

Reviewer #4

(Remarks to the Author)

The authors have revised their manuscript satisfactorily.

Version 2:

Reviewer comments:

Reviewer #2

(Remarks to the Author)

The authors have addressed all prior concerns.

Reviewer #3

(Remarks to the Author)

Thank you for the revisions made. I am satisfied with the authors response to previous comments.

Response to Reviewers' Comments

The authors deeply appreciate the reviewers' insightful and careful comments, which have greatly enhanced this manuscript.

In the annotated manuscript, the changes of text are tracked. Response to each reviewer's comment is marked by annotations to the text.

Reviewer #1 (Remarks to the Author):

The manuscript titled "[Enhancing Tacrolimus Dosing Precision in Liver Transplantation A Phenotypic Personalized Medicine Approach]" is an important contribution to the field of tacrolimus therapy for transplant patients, as it explores the use of phenotypic personalized medicine to optimize dosing and reduce the risk of toxicity or lower efficacy. The authors propose an easy-to-apply approach that has the potential to significantly improve outcomes for transplant recipients requiring tacrolimus immunosuppression. However, the manuscript would benefit from clarification of various statistical methods, more precise terminology, and correction of some inappropriate or unclear statements as follows (ordered based on the first appearance in the text not by significance):

1- Line 111: Was the number of patients with simultaneous liver and kidney transplantation higher in one arm than the other?

- As indicated in Table S1, there were five SLKT recipients in the SOC group and three in the PPM group.
- There were no statistically significant differences in LOS or percent days with a large deviation between the LT and the SLKT groups. The SLKT patient group had similar mean length of stay as the LT patient group, (14.9 days for SLKT vs. 15.9 days for LT, $p=0.47$) and also similar average percent days with a large deviation (33% for SLKT vs. 31% for LT, $p=0.68$).
- Separately analyzing the LT and SLKT groups does not change the overall findings; it only results in the SLKT group not having enough subjects to reach statistical significance. Please see the table (now included as Table S3) below for the details of the comparisons.
- Furthermore, it is important to recognize that this study was performed as a pragmatic clinical trial to test the real-world performance of PPM compared to SOC. The vast majority of transplant centers manage liver and liver-kidney recipients on the same service and these patients are cared for by the same clinical team. Consequently, the dosing for both types of recipients is performed by the same clinicians, which justified the inclusion of both groups in the same study.

	LT				SLKT			
	Both (n=48)	SOC (n=24)	PPM (n=24)	p	Both (n=8)	SOC (n=5)	PPM (n=3)	p
LOS (d)	15.7 (12.0)	18.0 (11.6)	13.4 (12.2)	0.011	14.4 (4.7)	16.6 (4.2)	10.7 (3.1)	0.07
% large deviation	31 (25)	41 (28)	22 (19)	0.014	33 (21)	26 (25)	43 (9)	0.3

- The response and the table are added as Table S3.

2- Line 161: Authors mentioned "TTL higher than the target range may cause neural toxicity and lower than the target range may lead toward organ rejection" This is not accurate as Tacrolimus toxicity is not only neurological but can be also in the form of acute nephrotoxicity.

- Thank you for noting the narrow depiction of tacrolimus toxicity. The sentence was edited to "TTL higher than the target range may cause neuro- or nephrotoxicity and lower than the target range may lead toward organ rejection." (Page 10, line 167)

3- Line 166: Authors mentioned "There was a significant difference between the two groups" A significant difference in what?

- Thank you for your close reading and request for clarification. We acknowledge that the original statement was not sufficiently specific in describing the significant difference observed between the two groups. The sentence has been revised to "There was a significant difference in the mean percentage of post-transplant days with large deviations from the TTL between the two groups." (Page 10, line 171)

4- Line 167: As the authors are providing numbers between brackets after the mean values, the statement "The mean percentage of post-transplant days with large deviations was 38.4 (27.4)%" needs to be changed to The mean percentage (standard deviation).

- Thank you for your close reading and request for correction. The edit has been made. (Page 10, line 173)

5- Line 168: Difference between 38.4-24.2 is 14.2% not 14.1%

- Thank you for your close reading and request for correction. The edit has been made. (Page 2, line 26, Page 3, Line 39, Page 10, line 173)

6- Lines 173-175: Authors mentioned "There were no tacrolimus level related episodes of nephrotoxicity or neurotoxicity and there were no graft failures or patient deaths within the one-year follow up of these patients." How these toxicities were assessed. Have you followed up the cases with serum creatinine levels, GFR, and urine outputs. Any statistical data for this? & For neurotoxicity have you followed this up with MRI?

- We appreciate the comments in understanding the methodology for assessing tacrolimus-related toxicities.
- Nephrotoxicity Assessment:
- Nephrotoxicity was monitored through regular assessment of renal function, including serum creatinine levels, glomerular filtration rate (GFR), and urine output. Specifically, serum creatinine levels were measured daily during the initial post-transplant hospitalization. GFR was estimated using the Modified Diet in Renal Disease (MDRD) equation. Urine outputs were also recorded daily during the initial post-transplant

hospital stay. These data were analyzed to identify any trends indicative of nephrotoxicity.

- Neurotoxicity Assessment:
- Neurotoxicity was evaluated via clinical assessments for neurologic symptoms, including changes in mental status, seizures, or other neurological deficits. Additionally, MRI imaging was utilized as appropriate for cases where clinical symptoms suggested possible neurotoxicity.
- Statistical Data:
- Regarding the incidence of nephrotoxicity, the following results were observed. Serum creatinine levels and GFR values did not show significant deviations from the baseline indicative of nephrotoxicity in either group. No statistically significant differences were noted between the SOC and PPM groups in terms of serum creatinine levels or estimated GFR at discharge.
- For neurotoxicity, MRI imaging followed any patient who exhibited clinical symptoms suggestive of neurotoxicity. In the present study, tacrolimus-induced neurotoxicity was assessed but not conclusively identified, as documented in the following instances. Two patients underwent MRI imaging, revealing no evidence of tacrolimus neurotoxicity (e.g., posterior reversible encephalopathy syndrome). (Edits were made to the results section - Page 11, lines 182-196; we also added further details in the supplementary section for how neuro and nephrotoxicity were assessed – page 33-34, lines 653-664)

7- Line 178: authors mentioned post-hoc test. What was the posthoc test used herein. Have you performed a normality test? If yes, results for the normality test are not provided.

- We appreciate the opportunity to clarify our terminology. In this context, we intended to refer to “exploratory” analyses rather than a specific post-hoc test. The terminology was changed throughout the text.

8-Line 184-185: Authors wrote this equation "Mean AUC OOR = sum [TTL(OOR)] / study dosed days" and the units of this AUC was ng/mL/day.

The equation you provided for calculating the AUC (area under the curve) outside of the therapeutic range needs to be rechecked as it does not accurately capture the concept of AUC. AUC typically represents the total drug exposure over time, calculated as the integral of the concentration-time curve. When considering AUC outside of a specified therapeutic range, the focus is on the drug concentrations that fall outside this range during the study period.

Sum of Trough Levels:

If you are summing trough levels (the lowest concentration of the drug in the bloodstream before the next dose), this may not accurately represent the AUC, which typically involves integrating all concentration values over time, not just trough levels.

Study Dosed Days:

Dividing by the number of study dosed days may provide an average concentration, but this does not equate to AUC. AUC is generally calculated as the area under the entire concentration-time curve, which requires more than just the trough levels.

To accurately calculate the AUC outside of the therapeutic range, consider the following steps:

Identify Concentration Levels: Collect all concentration data points (not just troughs) that fall outside the therapeutic range during the study period.

Calculate AUC: Use the trapezoidal rule or another integration method to calculate the AUC based on these concentration values over time. This will provide a more accurate representation of total exposure.

Revised Equation

A more accurate representation could be:

AUC outside of the range

$$= \int_{t_0}^{t_n} C(t) \cdot dt$$

Where

C(t) represents the concentration values outside the therapeutic range over the study period.

Conclusion

For a precise calculation, consider integrating all relevant concentration data over time, rather than relying solely on trough levels.

9- Line 185: ng/mL/day unit is not a unit of AUC and it is not similar to ng-day/mL. They are different units with different meanings:

- ng/mL/day: This represents a rate of change in concentration per unit of time

It would be interpreted as nanograms per milliliter per day. For example, a drug concentration increasing by 10 ng/mL each day. This unit does not represent an area under the curve (AUC)

- ng-day/mL:

This represents an area under the concentration-time curve (AUC). It would be interpreted as nanogram-days per milliliter. It is a valid unit for expressing AUC when concentration is in ng/mL

The time component (days) is multiplied by the concentration. Integrating this over the dosing interval gives the total exposure as explained above.

- For both comments 8 & 9:
- Thank you for your insightful observation regarding the use of the term AUC in our manuscript. We appreciate the opportunity to clarify our wording. It is correct that the terminology we used (i.e., AUC) does not accurately capture the concept we were trying to convey. Our intention was not to measure the AUC in the traditional pharmacokinetic sense.
- What we were measuring is the average daily deviation from the target range (TTL) rather than the true AUC as indicated by the reviewer. To reflect this accurately, we will change the terminology in the manuscript. Instead of "Mean AUC OOR," it should be termed "Average Daily Deviation Out-Of-Range (ADD)." The revised equation will be: "ADD OOR = sum [TTL(OOR)] / study dosed days". To clarify, OOR means TTL > 10 ng/mL or < 8 ng/mL if the clinical team has designated the therapeutic range to be 8-10.
- We emphasize that we were not intending to measure average concentrations or the actual AUC of tacrolimus concentrations. The study of tacrolimus AUCs and their predictability/variability is a complex endeavor and was not the goal of this particular study. Our focus was on quantifying how much daily TTLs deviated from the target range, expressed as an average over the study period.
- The term was changed throughout the text, including Page 12, lines 203-212

10- Figure 3 needs clarification. At what timepoint of the follow-up were these TTL data collected?

- We appreciate the reviewer's request for clarification. The TTL data presented in Figure 3 represent the mean TTLs averaged over the length of the study for each patient. We have added this information to the figure legend and clarified it in the corresponding section of the manuscript to avoid any ambiguity. (Pages 12-13, lines 207-212)

11- Line 196: The title "Early Recovery of AST Associated with Shorter Length of Stay of PPM Group" is not relevant to the text below it

- Thank you for noting that the section title was misplaced. We have fixed the error. (Page 15, line 229)

12- Line 210: The abbreviation (DAST) does not make sense with the description before it. Please check

- We thank the reviewer for identifying this lack of clarity. We have revised the text for clarity as follows: "To investigate the relationship between recovery from transplant-related liver injury and LOS, we analyzed the number of days (referred to as Days to AST Normalization or DAST), it took for AST levels (a standard measure of liver injury) to normal value, 48U/L." (Page 15, line 230)

13- Line 210: Authors mentioned that AST is used as it is the standard measure for liver injury. However, in clinical practice, alanine aminotransferase (ALT) is often prioritized when evaluating liver function and injury and is generally considered a better standard than aspartate aminotransferase (AST). Here are the key points regarding their roles and significance:

ALT vs. AST in Liver Injury

ALT Specificity: ALT is primarily found in the liver and is more specific to hepatocellular injury. Elevated levels of ALT are typically associated with different liver conditions such as viral, alcoholic, and non-alcoholic liver disease. In most liver injuries, ALT levels rise more significantly than AST levels, making it a more reliable marker for liver damage.

AST Sources: While AST is also present in the liver, it is found in other tissues, including the heart and muscles. Therefore, elevated AST levels can arise from non-liver sources, such as muscle injury or cardiac events, which can complicate the interpretation of liver function tests.

- We appreciate the reviewer's insightful comment regarding the use of ALT versus AST in evaluating liver injury. While it is true that ALT is more liver-specific and often prioritized in clinical practice, we opted to use AST in our study for the following reasons:
- AST has a shorter half-life (~17 hours) compared to ALT (~47 hours). This shorter half-life makes AST a more dynamic marker in the early post-transplant period, increasing the likelihood that AST levels would normalize prior to discharge. In this patient population, as noted in the text, 6 subjects did not reach normal AST levels; there were 30 patients whose ALT levels did not reach normal levels prior to discharge, making this measure unreliable. As a result, measuring the days to AST normalization (DAST) provided a more practical metric for assessing recovery from transplant-related liver injury in the immediate postoperative period.

- We agree with the reviewer that ALT is more liver-specific. However, in the context of early post-liver transplantation, we found that AST and ALT levels were highly correlated. This correlation is supported by the low incidence of muscle injury or cardiac events severe enough to disproportionately elevate AST over ALT. Therefore, in this specific clinical setting, AST served as a reliable and practical marker for liver injury recovery.
- Moreover, we did compare the Days to ALT Normalization (DALT) between the two groups and found that DALT was also lower in the PPM group than in the SOC group, but given the number of subjects who did not reach normal ALT levels this difference did not reach statistical significance (12.7 days for SOC vs. 8.0 days for PPM, $p=0.15$). This further supports the conclusion that both AST and ALT recovery were more rapid in the PPM group, reinforcing the validity of our findings. These results were added to the text (Page 16, lines 242-246)

14- Have you considered other biochemical tests such as: Serum Creatinine, or better Cystatin C (for kidney function), Proteinuria, Electrolytes, Donor-Specific Antibodies (DSA; immunological marker for rejection), Neutrophil Gelatinase-Associated Lipocalin (NGAL), Cell-Free DNA, or Urinary Chemokines such as CXCL9 and CXCL10?

- We appreciate the reviewer's suggestion to consider these additional biochemical tests. This trial was conducted as a pragmatic clinical trial with the objective of evaluating the performance of PPM in routine clinical practice of the daily dosing of tacrolimus to achieve a predetermined desired trough range. The biochemical tests mentioned by the reviewer, while potentially informative, are not part of the standard of care in the immediate post-liver transplant period. Furthermore, these tests are less relevant for the primary focus of this study, which was centered on improving drug dosing.
- It should additionally be noted that the cited biomarkers generally do not exhibit significant variability or provide actionable information over the short course of the index hospital stay following liver transplantation. The inclusion of these tests would not have contributed meaningful data to the primary endpoints of this trial, which were based on standard postoperative monitoring protocols.

15- Lines 247-248: what was the duration of follow-up for the primary outcome?

- Thank you for your comment. We have clarified the duration of follow-up for the primary outcome in the text. Specifically, the primary outcome was measured "during the initial post-transplant hospitalization." This duration encompasses the entire period from the time of first study-dosed tacrolimus after liver transplantation until the patient's discharge from the hospital. We have updated the manuscript to clearly reflect this timeframe for the primary outcome.
- Edits have been made to ensure this information is consistently stated throughout the relevant sections of the text. (Page 10, line 166)

16- Tables S1 & S2: Black subjects are more in the SOC than PPM that can contribute to the differences between the two groups apart from the personalized medicine initiative. Metabolism

by CYP3A5 is known to be higher in black race compared to whites.

Additionally, the prognosis of tacrolimus therapy can be worse in Black patients compared to White patients, primarily due to pharmacokinetic differences, dosing requirements, and potential socioeconomic factors. Here are the key points regarding this disparity:

- **Pharmacokinetics:** Studies indicate that Black kidney transplant recipients often have a faster clearance rate of tacrolimus compared to White recipients. This means they may require higher doses to achieve therapeutic levels, which can complicate management and increase the risk of both toxicity and rejection if not properly monitored.

- **Dosing Differences:** Black patients typically receive nearly 55% higher doses of tacrolimus than their White counterparts to achieve similar drug exposure levels. Despite higher dosing, achieving target concentrations can still be challenging, leading to a higher incidence of underexposure (i.e., lower than target trough levels) in Black patients which we see in SOC group.

- **Adverse Effects:** There is evidence that Black patients may experience more adverse effects from tacrolimus therapy, which can further complicate treatment outcomes. The cumulative effects, particularly in Black women, have been noted to be more pronounced.

- **Chronic Allograft Survival:** Research has shown that chronic renal allograft survival tends to be shorter in Black recipients compared to other racial groups receiving similar immunosuppressive regimens. Factors contributing to this disparity include medication adherence, genetic differences in drug metabolism (particularly related to the CYP3A5 enzyme), and socio-economic factors that can affect access to healthcare and medication management.

- **Socioeconomic Factors:** Disparities in healthcare access, socioeconomic status, and potential differences in health literacy may also contribute to the poorer prognosis observed in Black patients. These factors can influence medication adherence and overall health management, impacting transplant outcomes.

- We appreciate the reviewer's detailed and thoughtful comments on the potential impact of race on tacrolimus therapy, and we are pleased to provide a thorough response addressing each of the points raised.
- **Pharmacokinetics (CYP3A5 and Tacrolimus Clearance):** It is well established that Black patients, particularly those expressing the CYP3A5 *1 allele, tend to have faster clearance rates of tacrolimus. However, there were only two Black subjects in the SOC group and none in the PPM group. It is unlikely that these two individuals alone accounted for the differences observed between the groups. In clinical practice, the pharmacokinetics of tacrolimus, including any racial differences in metabolism, are typically managed by adjusting doses to achieve target trough levels. Importantly, in our study, clinicians were not blinded to race/ethnicity and could adjust tacrolimus doses based on clinical need. This would mitigate any potential racial disparity in tacrolimus pharmacokinetics.
- **Dosing Differences:** It is true that Black patients may require higher doses of tacrolimus to achieve target therapeutic levels, often due to the increased expression of functional CYP3A5. However, the dosing clinicians in our trial were aware of patient race/ethnicity and would have adjusted tacrolimus doses accordingly. The use of PPM is designed to optimize dosing based on individual patient response, independent of race. Since PPM focuses on real-time clinical data rather than population-based assumptions (e.g., race or ethnicity), it is race-neutral and agnostic to these differences. If race were a significant factor in determining tacrolimus dosing, PPM would have been expected to perform

worse than SOC. The fact that it did not suggests that PPM successfully managed these variations.

- **Adverse Effects:** The literature suggests that Black patients may experience more adverse effects from tacrolimus therapy. However, our study did not show a higher incidence of adverse effects, whether related to race or not, in the SOC group, where two Black patients were included. This could be due to the individualized dosing adjustments made, taking into account patient responses and potential side effects, or it could be that these adverse effects take a longer period of time to manifest than just the post-transplant hospital stay. Moreover, given the small number of Black subjects in the SOC group, the impact of any potential race-related adverse effects on overall outcomes would be minimal.
- **Chronic Allograft Survival:** Chronic allograft survival has been reported to be shorter in Black patients, likely due to a combination of factors including pharmacokinetic variability and socioeconomic factors. However, chronic allograft survival was not an endpoint in our study, which focused on the early postoperative period. While long-term outcomes such as allograft survival are important, they were beyond the scope of this trial. Furthermore, the personalized management approach used in PPM aims to address individual variations in tacrolimus dosing, which could potentially improve long-term outcomes across different racial and ethnic groups, but further studies would be needed to confirm this.
- **Socioeconomic Factors:** We agree that socioeconomic factors such as healthcare access, health literacy, and medication adherence can affect transplant outcomes, particularly in minority populations. However, in the controlled environment of a clinical trial, particularly one where the entirety of the trial was conducted in the hospital where all patients have access to standardized care and medication management, these factors are less likely to introduce significant variability. All patients, regardless of race or socioeconomic status, were closely monitored for medication adherence and response, and dosing was adjusted based on clinical need rather than socioeconomic factors.
- In summary, while we recognize that race-related pharmacokinetic differences and socioeconomic factors can influence tacrolimus therapy outcomes in clinical practice, the small number of Black subjects in our study makes it unlikely that these factors had a meaningful impact on the overall results. The PPM approach used in our trial is race-agnostic, relying on real-time therapeutic drug monitoring to optimize tacrolimus dosing, and thus effectively managed variability in tacrolimus pharmacokinetics, including those potentially influenced by race. Edits were made in the discussion to acknowledge this (page 19, Line 301).
- Nevertheless, we analyzed the primary outcome separately for the Black and Non-Black subjects.
- As indicated in Table S1, there were two Black recipients in the SOC group and none in the PPM group.
- Separately analyzing the groups does not change the overall findings; it only results in the Black recipient group not having enough subjects to perform any statistical analysis on. Please see the table below (added as Table S3, Page 30) for the details of the comparisons.
- Furthermore, it is important to recognize that this study was performed as a pragmatic clinical trial to test the real-world performance of PPM compared to SOC. The vast majority of transplant centers manage liver and liver-kidney recipients on the same

service and these patients are cared for by the same clinical team. Consequently, the dosing for both types of recipients is performed by the same clinicians, which justified the inclusion of both groups in the same study.

	Non-Black				Black		
	Both (n=54)	SOC (n=27)	PPM (n=27)	p	SOC (n=2)	PPM (n=0)	p
LOS (d)	15.6 (11.4)	18.1 (10.9)	13.1 (11.5)	0.002	33.3 (9.4)	-	-
% large deviation	31.5 (25)	38.8 (28.4)	24.3 (19.1)	0.033	13.5 (6.4)	-	-

•

Reviewer #2 (Remarks to the Author):

In this single-center, pragmatic, randomized, partially blinded trial, liver transplant patients were randomized to undergo daily standard-of-care (SOC) physician-guided dosing or Phenotypic Personalized Medicine-PPM guided dosing of tacrolimus (PPM platform uses quantitative phenotypic data -Tacrolimus trough levels-TTL in the prior 3 days) to personalize treatment. The main end point was the percent days with large (>2 ng/mL) deviation from target range during the initial post-transplant hospital stay. The main finding was the statistically significant improvement in the PPM group: 24.3% of days showing large deviations compared to 38.4% in the SOC group (difference -14.1%, 95% CI: -26.7 to -1.5 %, P=0.029).

Ensuring therapeutic target levels is very important for transplant recipients yet a simple automated way has been developed yet due the enormous complexity of the LT system and tac pharmacokinetics. Any improvement in that sense is welcomed.

There are some aspects that need to be addressed:

1. The study was randomized yet the minimum number of patients that needed to be reached to achieve conclusive evidence (30 per arm) was not reached once some patients were discarded from the analysis. 27 and 29 patients in each arm. Were all consecutive patients included in the study in this "large transplant center" during a 1.9 years period? How many patients were not included during that period and what were the reasons?

- We included all consecutive patients who received a liver organ offer between September 1, 2018, and June 4, 2020, at our center for this study, as outlined in Table 1 (the CONSORT Flow diagram), with no exclusions from randomization. The publicly accessible Organ Procurement and Transplantation Network (OPTN) database (<https://optn.transplant.hrsa.gov/data/view-data-reports/center-data/>) reflects that our center conducted 30, 30, and 84 liver transplant operations in the years 2018, 2019, and 2020, respectively, when the study was being conducted aligning with the enrollment figures. It is worth noting that the COVID-19 pandemic did impact the number of transplants performed in 2020. We have removed the word large to better reflect the liver transplant volume at the time of the study. (Page 7, line 112)

2. The study was not completely blinded such that subtle changes in the management of the SOC or PPM group can not be discarded. Were the physicians in charge of the patients the same during the study period? what was the overall LT experience of these physicians? was it similar for both groups?

- Thank you for identifying the potential for subtle management differences between the groups, which is an important consideration.
- The standard practice at our center, as it is in most transplant centers worldwide, relies upon the care team on call to determine the immunosuppression dosing. This study was intentionally designed as a pragmatic clinical trial to compare real-world dosing of tacrolimus with the PPM approach.
- **Physicians in Charge:** Over the two years of the study, the team providing care to the patients remained the same. This team was comprised of a transplant pharmacist with 5 years of experience at the start of the study and four transplant surgeons with between 5 and 20 years of experience in liver transplantation.
- **Experience Equivalence:** The transplant pharmacist and surgeons managed both groups, ensuring that the clinical expertise and judgement applied were consistent regardless of the study arm.
- **Blinding and Dosing:** While the PPM group was NOT dosed by a transplant clinician per se, the recommended dose generated by the PPM algorithm was reviewed for safety by the on-call clinician before being ordered. Importantly, none of the PPM recommendations were overridden, as noted in the manuscript. The entire remainder of patient care, including monitoring, diagnostics, and supportive care, was conducted according to standard-of-care practices, ensuring no bias was introduced in other aspects of patient management.
- Edits were made throughout the text to clarify these points.

3. The authors state that the end-point was the % of large deviations during the initial post-transplant period? What do they mean by Initial? is initial the same by group? if LOS different in each group, the number of measurements are not the same by group and this may lead to bias. Were the number of measurements performed by group similar?

- Thank you for highlighting the need for clarity regarding the "initial" post-transplant period and its implications for measurement and potential bias. We have revised the language in the manuscript to specify that the "initial" post-transplant period refers to the duration from three days after the first tacrolimus dose post-liver transplantation until the day of hospital discharge (Page 10, line 170). This means that the days included in calculating the primary endpoint for both groups begin after this initial 3-day period and continue until discharge.
- **Addressing LOS Differences:** It is true that the LOS differs between groups, and this could theoretically introduce bias due to varying numbers of measurements. However, longer LOS typically results in easier dosing as dosing becomes more predictable with time since transplantation. To mitigate potential bias, we performed an additional

analysis using a cutoff date for calculating the primary endpoint at 10 days post-transplantation, thus limiting the effect of different LOS.

- In this truncated analysis, the mean percentage (standard deviation) of post-transplant days with large deviations was 42.1 (30.9)% in the SOC group and 24.9 (19.5)% in the PPM group (difference -17.2%, 95% CI: -31.1 to -3.5%, P=0.015).
- This analysis demonstrates that the significant difference between the groups is *even larger* (17.2% versus 14.1%) when controlling for LOS differences, indicating that our findings are robust.
- Corresponding edits were made on Page 10, lines 177-181

4. The authors state: "Large deviations (> 2.0 ng/mL) from trough target ranges were deemed especially important as they were correlates of possible adverse events". Why was 2 and not 1 or 3 considered as relevant. Clearly this relates to the baseline /previous trough level. Changing from 1.5 ng/ml to 3.6 has clearly not the same meaning as changing from 10 to 12.1 or from 4 to 1.9. Time from LT is also important, situation of the patient, concomitant IS therapy. All are very relevant aspects that need to be considered instead of applying a "specific 8-10 range for all?". and this range may even change depending on the condition of the patient, particularly development of serious infections. How is this incorporated in the equation?

- Thank you for your observation regarding the selection and implications of the 2 ng/mL deviation threshold. This primary outcome was selected based on our pilot studies previously published in *Science Translational Medicine*. The study was powered based on those findings. The choice of 2 ng/mL for defining large deviations was informed by our previous pilot studies that identified this threshold as a significant correlate of adverse events. Additionally, discussions with transplant clinicians guided this decision, establishing 2 ng/mL as an appropriate cutoff for what would constitute a large deviation versus a small deviation. This threshold was chosen based on clinical judgement regarding what deviations would be most concerning in practice.
- **Standard of Care and Target Ranges:** The center's standard of care has established TTLs spanning a 2 ng/mL range at various periods post-transplantation, as follows:
 - 8-10 ng/mL for the first month,
 - 6-8 ng/mL for months 1-4,
 - 4-6 ng/mL thereafter.
- The goal of the present study was not to test the efficacy of these target ranges but rather to explore the ability to achieve them consistently, regardless of its specific value at a given time.
- **Clinical Context and Variability:** While the majority of the time the target TTL was 8-10 ng/mL, there were instances where clinical necessity dictated different ranges. It is crucial to emphasize that the determination of the target TTL was made by the clinical team per center standard of care, not by the study team. Specifically, for 4 subjects in the SOC arm and 5 subjects in the PPM arm, the target was adjusted during their initial hospital stay based on clinical considerations, including etiology of liver disease, infection status, and kidney function.
- **Incorporation of Clinical Factors:** We did not make a qualitative distinction regarding whether the difference between 1.5 ng/mL and 3.6 ng/mL was more or less meaningful than that between 10 ng/mL and 12.1 ng/mL. The goal was simply to achieve the TTL within the predesignated clinically indicated range. The PPM algorithm only used

previous TTLs and previous tacrolimus doses to arrive at the dose recommendation, without direct incorporation of additional clinical factors such as time from transplantation, patient condition, or other concomitant therapies. This simplified approach focused on individualized dose-response relationships to predict and achieve target TTLs efficiently.

5. The authors state that "No significant differences arose by chance in the baseline characteristics of the two groups. ". Yet some % are quite different. For instance, 26% of PPM group was HCC vs 10% in SOC group. Generally speaking HCC patients are easier to manage and target levels are more easily and rapidly achieved. Given the small number of patients included in this study, these differences between groups may have had an impact on the overall result.

- Thank you for your observation regarding the differences in baseline characteristics, particularly the higher percentage of HCC patients in the PPM group. As indicated in Table S1, there were three recipients with HCC in the SOC group and seven in the PPM group. This difference was not statistically significant (p=0.17).
- Nevertheless, to address this concern, we separately analyzed the groups with and without HCC. Our findings showed that there were no statistically significant differences in LOS or percentage of days with large deviations between the HCC and No-HCC groups. The HCC patient group had a comparable mean length of stay as the No HCC patient group, (14.2 days for HCC vs. 15.8 days for No HCC, p=0.68). Similarly, the HCC patient group had an average percentage of 32.9% of days with a large deviation compared to 25.3% in the No HCC group (p=0.43). While not statistically significant, the HCC subjects tended to have more deviations than the non HCC subjects, which goes counter to the assertion that HCC patients are easier to manage and target levels are more easily and rapidly achieved.
- **Impact on Overall Findings:** Separately analyzing these groups does not change the overall findings; it only results in each of the groups not having enough subjects to reach statistical significance. Please see the table below (added as Table S4) for the details of the comparisons.
- **Study Design Justification:** Furthermore, it is important to recognize that this study was performed as a pragmatic clinical trial to test the real-world performance of PPM compared to SOC. The difficulty of dosing tacrolimus after liver transplantation primarily arises from rapidly changing physiology and patient heterogeneity rather any given disease etiology. Dosing for both types of recipients is performed by the same clinicians, which justified the inclusion of both groups in the same study.

	No HCC				HCC			
	Both (n=46)	SOC (n=26)	PPM (n=20)	p	Both (n=10)	SOC (n=3)	PPM (n=7)	p
LOS (d)	15.9 (12.0)	17.8 (10.9)	13.2 (11.8)	0.18	14.9 (5)	17.3 (9.7)	12.9 (11.6)	0.56

% large deviation	31 (25)	38.3 (27.3)	26.0 (17.4)	0.068	33 (21)	38.9 (34.7)	19.5 (24.4)	0.45
-------------------	------------	-------------	----------------	-------	------------	----------------	-------------	------

-

6. The explanation of the PPM system could be improved. Does that system only consider TTL during the previous 3 days or does it incorporate other variables such as level of ALT elevation, or donor age?

- We appreciate the opportunity to clarify PPM. PPM as presented in the study uses the tacrolimus trough levels (TTL) from the previous day and the tacrolimus dose administered on that day to determine the next day's dose. Specifically, the PPM dosing equation (Function 2) is based on the tacrolimus dose for the next day, $c(t+1)$, calculated as follows: $c(t+1)=[TTL(t+1)/TTL(t)] \times c(t)$ where $TTL(t+1)$ is the desired next-day TTL within the therapeutic window. It does not directly incorporate additional clinical variables such as lab levels, donor age, concomitant medications, or any other patient characteristics. Instead, it focuses on the relationship between the prior TTL, the subsequent TTL, and the previous dose to dynamically adjust the tacrolimus dosing.
- We have updated the manuscript to clarify this point. (Page 6, lines 88-99)

7. While there was a statistical difference in the number of days showing large deviations, there were no differences in mean AUC above target range between the two groups. And while there were

differences related to underdosing (SOC group had a statistically significantly larger mean AUC below target range than the PPM group), this did not impact rejection episodes or graft loss? In fact, the differences using this PPM approach did not translate into clinical differences (same rate of rejection, graft loss, episodes of neuro or nephrotoxicity).

In essence, statistical differences but clinically relevant? In those with large deviations, it is important to understand the degree of deviation? What was the mean deviation in one group over another? How many patients did it involve? given the small sample size a more detailed analysis would be welcomed.

- Thank you for your comments regarding the clinical relevance of our findings.
- **Study Time Period and Event Frequency:** We acknowledge that the study period was short, focusing only on the immediate post-transplant period. This timeframe can preclude infrequent adverse events such as rejection, graft loss, and episodes of neuro or nephrotoxicity. Consequently, the study was not powered to detect differences in these clinical outcomes within this short duration.
- **Clinically Relevant Differences:** Despite the limitations in detecting the listed clinical outcomes above within the study period, we do in fact observe a clinically relevant difference consistently noted both in our pilot study and this study: the length of stay. This outcome is critical as it reflects overall patient stability and resource utilization.
- **Degree of Deviation:** Regarding the degree of deviation, Figure 4 in the manuscript presents individual subject data for both overdosing and underdosing. To address your request for further detailed analysis, we have now included a new Figure 4C showing the

overall average daily deviation for each group. This additional figure provides a clear visualization of the extent of deviations and permits a more detailed comparison.

Other comments

8. Table S2: the differences between the two randomized groups are shown for some but not all variables. For instance, HCC? MELD? Donor?

- All the factors listed in Table S1 are now included in Table S2

9. Please add some comments related to the lack of data when using prolonged release TAC, used in many LT centers in the world, which has shown better clinical results than standard twice daily tac.

- Thank you for your recommendation regarding the use of extended-release tacrolimus in liver transplantation. The standard of care at this center uses twice-daily tacrolimus. This approach is consistent with protocols at most transplant centers nationwide, particularly during the immediate post-transplant period, for several reasons:
- **Immediate Post-Transplant Period:** The extended-release formulation of tacrolimus does not have an FDA indication for use in liver transplantation, making the twice-daily formulation the preferred option, especially soon after transplantation. This period involves significant variability in dosing requirements and thus many dose adjustments need to be made, as noted in the text. These dose adjustments are impeded by the extended-release pharmacokinetics
- **Clinical Outcomes:** While extended-release tacrolimus has shown better clinical results in some settings, it is not definitively established that it leads to better outcomes in liver transplantation as compared to the twice-daily regimen (<https://www.ncbi.nlm.nih.gov/pmc/articles/PMC8006077/>).
- **Pharmacokinetics:** The pharmacokinetics of extended-release tacrolimus necessitate dose adjustment over a longer timescale, which may not be feasible in this short post-transplant inpatient setting where frequent adjustments are required. The dosing changes that occur during the immediate post-transplant period would require delays in dose optimization with extended-release formulations.
- **Broader Implications of PPM:** This study also demonstrates the power of PPM in settings where dosing is difficult and crucial, such as anticoagulation or chemotherapy. While specific formulations like extended-release can provide solutions to individual problems, the broader application and efficacy of the PPM approach offer significant advantages in personalizing and optimizing drug regimens across various complex medical conditions.
- Edits were made to the limitations section to acknowledge the lack of data for the extended-release formulation. (Page 19 – line 314)

10. The authors state that "Appropriate immunosuppression levels, i.e., keeping patients within

the higher in-range TTL using PPM (Figure 3), help in the liver's recovery and function, as evidenced by a quicker return to normal AST levels". Yet it could be the opposite: a better and more rapid return to NO liver damage leads to lesser TAC variability.

- Thank you for your comment regarding the relationship between liver function recovery and TTL variability. It is true that a better and more rapid return to normal liver function can lead to less TTL variability. However, it is important to highlight that this study was conducted as a randomized clinical trial, where the primary difference between the SOC and PPM groups was the dosing approach. Randomization helps ensure that any other potential differences between the groups are minimized. If there were an inherent reason for one group to exhibit less liver damage unrelated to the dosing method, we would expect to see significant differences in baseline characteristics or known factors affecting liver damage and TTL variability between the two groups. However, this was not the case in our study. Given the lack of other significant factors contributing to differences in liver function recovery, we can reasonably conclude that the observed differences in liver recovery and TTL variability between the SOC and PPM groups are attributable to the dosing approach utilized by the PPM system.

11. Is there data from the kidney transplant setting? some patients were both Liver and Kidney transplant recipients? I understand the numbers are small, but what were the specific results in that subgroup?

- Thank you for your question regarding the data from the kidney transplant setting and the outcomes for patients who received simultaneous liver and kidney transplants.
- Please see the response to Reviewer 1, Comment 1, for more details on this matter.

Reviewer #3 (Remarks to the Author):

This article is to describe the conduct and resulting data and outcomes of a randomised trial to compare a novel personalised method for deciding dosing of tacrolimus following liver and/or kidney transplantation compared to standard of care.

Overall it provides a good succinct description of the RCT performed. There are a number of recommendations which could improve the readers understanding of the research and evaluate it's likely value and impact.

1. Comment, line 128 states "study ended when the planned number of subjects were enrolled and completed the study" I suggest removing the latter part or rephrasing as the statement isn't quite right. The planned size was 30 per group, which although this number was enrolled, they didn't all complete the trial satisfactorily as stated in the paragraph above.

- Thank you for your suggestion. We have revised the sentence for accuracy to "study ended when the planned number of subjects were enrolled." (Page 7, Line 129)

2. Table 1: It seems a little unusual to only have screened the exact number that were found eligible and consented to randomisation in the trial. In a consort flow diagram you would expect the study team to have screened and approached several more patients than those that ended up being included as you find ineligible patients or on explaining the study to the patient they fail to provide their consent or are reluctant to take part in research and therefore are never included. The number screened in this case is the same as those randomised. Furthermore in line 110, the authors state that “Between September 1, 2018, and June 4, 2020, all adults scheduled to undergo primary, or redo liver or simultaneous liver/kidney transplantation at a large academic transplant center were screened for eligibility.” again suggesting that more patients might have been screened and not all would be expected to take part. It is important to describe screen failures to evaluate possible selection bias in the recruitment strategy as a single site study.

- Thank you for your observation and for highlighting the importance of detailing the screening and recruitment process. Please also see the response to Reviewer 2, Comment 1, which brings up a very similar point. In summary: As outlined in Table 1, the CONSORT flow diagram shows that there were no exclusions from randomization. Our patient population is analogous to many other transplant recipient populations characterized by high engagement in the process and strong relationships with the transplant team. During the study period, all adults scheduled for primary, redo liver, or simultaneous liver/kidney transplantation were screened for eligibility, and all approached patients consented to participate in the study. This high participation rate is a reflection of the strong rapport and trust between the transplant team and the patients, which likely facilitated consent. This aspect of our recruitment process means there were no screen failures, hence the identical number of patients screened and randomized.

3. Randomisation: It should perhaps be acknowledged that the randomisation sequence was generated by the PI using fixed block sizes and therefore cannot accurately be considered independent or be seen to maintain concealment of allocation as is customary in RCTs.

- We appreciate your concerns about the randomization process and the potential impact on allocation concealment. We will add an acknowledgment in the manuscript in the limitations section (Page 19 – line 311) to address this point.

4. Analysis plan: There is no mention of an analysis plan detailing how withdrawals or losses of data would be dealt with prior to data lock and unblinding of allocation in order to conduct analyses. Not including all randomised participants in a true intention to treat principled analysis plan may have led to the introduction of bias. An ITT analysis was reported as a secondary analysis, however there isn't a methods section describing how this analysis population was defined or derived making it hard to judge the importance/significance.

- Thank you for your valuable feedback regarding detailing the analysis plan and the need for clarity on how withdrawals or losses of data were handled. Prior to data lock and unblinding, we followed a predefined protocol for handling such occurrences. All randomized participants were included in the analysis for the days that were study dosed. Withdrawals and losses of data were documented, with reasons recorded. The primary analysis was performed on participants who completed the study as planned. TTLs that were not study dosed were excluded from analysis. Additionally, we performed, an ITT analysis as a secondary analysis to assess the robustness of the findings. The ITT population included all randomized participants based on their initial group assignments. However, there were subjects, such as those who did not undergo liver transplantation after randomization due to the cases being aborted (n=2), who could not be included in the ITT analysis as they did not have any measured TTLs. We have slightly adjusted the description of these 2 patients in the enrollment description to make it clearer (Page 7, Line 116-127). We have also revised the text in the Methods Section (Page 22, Line 382-388) to reflect this.

5. Line 37: refers to no significant differences between groups, these tables are presented as supplemental tables. Could the authors quantify how this assertion was made, did statistical tests confirm this, if so these results should appear in the S1 and S2 tables.

- Thank you for your comment. It is standard practice in RCTs not to report p-values for baseline characteristics if no significant differences between groups are observed, as any differences are expected to occur by chance. In our study, we followed this convention and did not include p-values in these tables. Below are the Supplementary Tables S1 and S2 to include the p-values from statistical tests performed for each baseline comparison, confirming that there were no statistically significant differences between the groups.

-
- Table S1

	Standard of Care (n=29)	PPM (n=27)	p-value
Male	16 (55%)	16 (59%)	0.75
Female	13 (45%)	11 (41%)	
HCC	3 (10%)	7 (26%)	0.17
SLKT	5 (17%)	3 (11%)	0.71
Redo OLT	1 (3%)	1 (4%)	1.0
Recipient Race/Ethnicity			0.14
Non-Hispanic White	26 (90%)	23 (85%)	
Hispanic White	1 (3%)	4 (15%)	
Non-Hispanic Black	2 (6%)	0	
Recipient Age (years)	57 (41-61)	58 (51-64)	0.47
BMI (kg/m ²)	27.9 (25.4-37.5)	28.6 (23.3-32.6)	0.98
NaMELD	27 (18.5-30.5)	27 (15-32)	0.67

Warm Ischemia Time (min)	31 (26.5-38.5)	33 (26-39)	0.84
Cold Ischemia Time (min)	390 (310-440)	330 (300-420)	0.44
DCD Donor	0 (0%)	1 (4%)	0.48
Donor Age (years)	41 (25.5-58)	38 (32-54)	0.84
Hepatitis C Positive Donor	1 (3%)	2 (7%)	0.60
Donor Risk Index	1.38 (1.25-2.01)	1.35 (1.07-1.66)	0.18
Dialysis After Transplant	3 (10%)	2 (7%)	1.0
Donor Cause of Death			
Anoxia	8 (28%)	11 (41%)	0.48
Cerebrovascular Accident	11 (38%)	10 (37%)	
Trauma	10 (34%)	6 (22.2%)	
Fluconazole Use After Transplant	17 (59%)	16 (59%)	1.0
Mycophenolic Acid Use	2 (7%)	2 (7%)	1.0
Basiliximab Use	10 (34%)	7 (25.9%)	0.57
Tacrolimus Target Range Other Than 8-10 ng/mL	4 (14%)	5 (19%)	0.72

- Table S2

	Standard of Care (n=31)	PPM (n=31)	p-value
Male	17 (55%)	18 (58%)	1.0
Female	14 (45%)	13 (42%)	
HCC	3 (10%)	8 (26%)	0.18
SLKT	6 (19%)	3 (10%)	0.47
Redo OLT	1 (3%)	2 (6%)	1.0
Recipient Race/Ethnicity			0.30
Non-Hispanic White	27 (87%)	26 (84%)	
Hispanic White	2 (6%)	4 (13%)	
Non-Hispanic Black	2 (6%)	0	
Asian	0	1 (3%)	
Recipient Age	57 (42-61)	58 (50-63)	0.62
BMI (kg/m ²)	27.7 (24.9-32)	28.6 (22.7-32.6)	0.99
NaMELD	27 (19-29)	27 (15-32)	0.66
Warm Ischemia Time (min)	32 (27-37)	33 (26.5-37.5)	0.78
Cold Ischemia Time (min)	390 (314-440)	333 (301.5-423.5)	0.49
DCD Donor	0 (0%)	1 (4%)	0.49
Donor Age (years)	37 (25-57)	39 (32-54)	0.89
Hepatitis C Positive Donor	2 (6%)	2 (6%)	1.0
Donor Risk Index	1.38 (1.22-1.73)	1.35 (1.18-1.66)	0.45
Dialysis After Transplant	4 (13%)	2 (6%)	0.67
Donor Cause of Death			0.44
Anoxia	9 (29%)	11 (38%)	
Cerebrovascular Accident	11 (35%)	12 (41%)	
Trauma	11 (35%)	6 (21%)	
Fluconazole Use After Transplant	19 (61%)	16 (55%)	0.79
Mycophenolic Acid Use	2 (6%)	2 (6%)	1.0
Basiliximab Use	11 (35%)	8 (25.8%)	0.58
Tacrolimus Target Range Other Than 8-10 ng/mL	4 (13%)	5 (16%)	0.73

6. Line 166 simply states “There was a significant difference between the two groups.” there is no indication what difference this is referring to (the previous or next sentence for example) and should be stated clearly and precisely within that sentence or removed completely.

- Thank you for noting this lack of clarity in the language. The sentence was revised. (Page 10, Line 171)

7. The result of the secondary outcomes, fraction of days outside of target range doesn't appear to be reported at all and should included for transparency since it was an endpoint stated in the protocol.

- Thank you for noting the absence of this outcome in the results. The results are now included on page 11, Line 185-199.

8. Line 173 states no safety endpoints of nephrotoxicity and neurotoxicity related to tacrolimus where observed. Maybe it can be clarified if these events were observed but not considered related to tacrolimus, or that no events occurred at all in either arm? If the former the number and difference between groups should be reported in each group since relatedness could be subjective in an open label study.

- Thank you for noting the absence of these outcomes in the results. The results are now included on page 11, Line 192-199.

9. The analysis methods of the exploratory DAST endpoint have not been stated, on line 212 it states “Six patients, three in each group, whose AST levels did not reach normal levels prior to discharge, were excluded from this analysis.” suggests that the DAST variable was considered as a binary outcome rather than time to event data that could have included the 6 participants that were removed from the analysis and therefore be considered more appropriate for comparisons purposes.

- Thank you for your insightful comment. To clarify, DAST was not treated as a binary variable. The analysis excluded the six patients whose AST levels did not return to normal by discharge because the study protocol concluded upon discharge, at which point all subjects transitioned to standard-of-care dosing (and beyond the control of the study). However, we acknowledge the potential for treating this as time-to-event data for every subject by including post-discharge days in the analysis. We have reanalyzed the data with DAST extending beyond the LOS and found that this adjustment does not

materially affect the results. We have included these findings in the revised manuscript for completeness. (Page 15, Line 238-240

10. The one sentence summary at the beginning of the article should perhaps reflect the primary endpoint results (as a minimum) in addition to or instead of the post hoc analysis of length-of-stay reported.

- Thank you for the suggestion. The one sentence summary was edited to include the primary endpoint. Page 2, Line 25-28

11. Overall the article would benefit from a methods section defining elements (as a minimum) such as how the different analysis populations were derived, the outcome measures analysed including how they were derived (especially those not stated up front in the protocol) and the analysis methods and test statistic used for each hypothesis test carried out and presented stating significance level and if 1 or 2-sided.

- Thank you for your suggestion regarding the need for a more detailed methods section. In response, we have expanded the statistical analysis section to provide clearer definitions of the analysis populations, outcome measures, and the corresponding statistical methods used. Specifically, we have clarified how the different analysis populations were derived and provided additional information on outcome measures not stated in the protocol but analyzed in the study. Additionally, we have specified the test statistics used for each hypothesis test, as well as the significance level. All tests were two-tailed. Page 22-23.

12. The discussion was sensible and outlined the main findings with the exception of the contradiction in results in terms of TTL outside of range. The stated secondary endpoint of fraction of days outside of target range was not presented. Furthermore, there were some inconsistencies in the results with the planned AUC OOR endpoint analysis providing a 'non-statistically-significant' result (p-value) whereas the apparent post-hoc analysis (i.e. not specified in the protocol a priori) of mean TTL did provide a 'statistically-significant' result (p-value). Perhaps the discussion might acknowledge this and discuss the potential for type I errors (false positive p-values) given the large number of hypothesis testing, post hoc analyses and apparent lack of adjustment for multiple testing.

- Thank you for your thoughtful feedback. We acknowledge the potential for inconsistencies between the AUC OOR (now termed ADD) and mean TTL results and the possibility of type I errors due to multiple hypothesis tests. However, we would like to emphasize that the primary endpoint—the percentage of days with large deviations from the target trough drug level—did show a statistically significant difference between the PPM and SOC groups. This was a pre-specified primary outcome and, as such, does

not require adjustment for multiple comparisons. We have revised the discussion to clarify this point and to highlight the distinction between the primary outcome and the exploratory analyses, which warrant more cautious interpretation due to the lack of adjustment for multiple testing. (Page 20, Line 319-325)

Response to Reviewers' Comments

The authors again appreciate the reviewers' insightful and careful comments.

In the annotated manuscript, the changes are tracked. Responses to the reviewer comments marked by annotations to the text.

Reviewer #2 (Remarks to the Author):

The authors have adequately addressed all comments.

One additional question that was not addressed is the cost: is it different?

- Thank you for your positive feedback. Regarding the additional question on the cost difference, we would like to clarify that the cost associated with the PPM approach is minimal. This is primarily because all the collected data necessary for the PPM approach are part of the standard of care procedures already in place. Furthermore, the processing and calculation required for PPM can be performed by a trained transplant clinician in approximately the same amount of time as is needed for standard dosing decisions. Therefore, the implementation of PPM does not result in significant additional costs or require extensive new resources.
 - The edits are made on Page 19, lines 310-314.
-

Reviewer #3 (Remarks to the Author):

I thank the authors for their thorough examination of the previous comments and for clear and concise responses and amendments to the manuscript.

- Thank you for your continued constructive feedback.

1 – There is one remaining point to be considered, in response to reviewer #3 point 4, the authors state:

“The primary analysis was performed on participants who completed the study as planned. TTLs that were not study dosed were excluded from analysis. Additionally, we performed, an ITT analysis as a secondary analysis to assess the robustness of the findings. The ITT population included all randomized participants based on their initial group assignments. However, there were subjects, such as those who did not undergo liver transplantation after randomization due to the cases being aborted (n=2), who could not be included in the ITT analysis as they did not have any measured TTLs. We have slightly adjusted the description of these 2 patients in the enrollment description to make it clearer (Page 7, Line 116-127). We have also revised the text in the Methods Section (Page 22, Line 382-388) to reflect this.”

The additions to the text are welcome and go some way to addressing the comment, thank you. Please note the typo in the additional text provided (page 22, line 387) the word 'who' is repeated and should be removed.

- Thank you for your close reading. We apologize for the typographical error identified in the text. We have corrected the typo by removing the repeated word "who". (Page 23, line 393)

2 – Use of the phrase ‘intention to treat’ is perhaps still misleading since it implies all participants are included in the analysis (true meaning of the phrase) whereas the authors clearly exclude some participants due to missing data. This approach might be more accurately described as a ‘modified intention to treat approach’. It is a limitation of the methods used that a true ITT population hasn’t been analysed whereby imputation for missing data could have enabled inclusion of all randomised participants.

I would suggest the authors ideally include the number of participants analysed in each of the analysis populations ‘completed study’ and ‘ITT’ in the consort diagram. Instead of the generic ‘analysed n=x’ it would be more transparent to list each analysis population and state the n that was analysed.

- Thank you for your insightful feedback. We acknowledge that the term "intention to treat" (ITT) may be misleading in this context. As for the suggestion to perform imputation for missing data, we acknowledge that this could be a potential method to include all randomized participants. However, given the limited data/subjects, we believe that imputations may not be robust. Therefore, we opted for the mITT approach to ensure the integrity of our findings. We have updated our terminology to "modified intention to treat" (mITT) to more accurately reflect our approach. Furthermore, we have revised the CONSORT diagram to clarify the number of participants analyzed in each group ('completed study' and 'mITT') for greater transparency.
- Changes made:
 - Revised Text in the Results Section: "This difference remains statistically significant with a modified intention-to-treat analysis (including all subjects except for two randomized to the PPM group who did not undergo liver transplantation)" (Page 10, lines 176-177)
 - Revised Text in the Methods Section: "In addition, a modified intention-to-treat (mITT) analysis was conducted as a secondary analysis to assess the robustness of the findings. The mITT population included all randomized participants based on their initial group assignments." (Page 23, lines 390-391)
 - Revised CONSORT Diagram: Revised Text: "Completed Study Population, n=x" and "Modified intention to treat analysis (including all subjects except for those not transplanted, n= x" (Page 8, line 131)

Reviewer #4 (Remarks to the Author):

The authors have revised their manuscript satisfactorily.

- Thank you for your positive feedback and for recognizing the revisions made to our manuscript. We are pleased that you found the updates satisfactory.